# Effector innovation in genome-reduced phytoplasmas and other host-dependent mollicutes

**Federico G. Mirkin**, **Sam T. Mugford**, **Vera Thole**¤, **Mar Marzo**, **Saskia A. Hogenhout**\*

Department of Crop Genetics, John Innes Centre, Norwich Research Park, Norwich, United Kingdom.

¤ Current address: Department of Biochemistry and Metabolism, John Innes Centre, Norwich Research Park, NR4 7UH, Norwich, UK
\* saskia.hogenhout@jic.ac.uk

## Abstract

Obligate host-associated bacteria with reduced genomes, such as phytoplasmas, face strong evolutionary constraints, including metabolic dependence on hosts, limited opportunities for horizontal gene transfer (HGT), and frequent population bottlenecks. Despite these limitations, phytoplasmas, which are parasitic, insect-transmitted plant pathogens, maintain a diverse arsenal of secreted effectors that manipulate both plant and insect hosts to promote infection and transmission. These effectors can suppress immunity and reprogram plant development, inducing alterations such as witch's broom and leaf-like flowers, through ubiquitin-independent degradation of key transcription factors. However, how phytoplasmas diversify and maintain these effectors in the absence of frequent genetic exchange remains unclear. To address this, we analysed the effectoromes of 239 phytoplasma genomes and identified a diverse set of secreted proteins, which we designated as putative Phytoplasma Effectors (PhAMEs). We found that PhAMEs targeting evolutionarily conserved and structurally constrained surfaces of host proteins are widespread across phytoplasmas. These effectors adopt compact, efficient folds. They often function as molecular scaffolds with dual interaction surfaces capable of linking host proteins or integrating signalling pathways. Such scaffolding PhAMEs have evolved multiple times independently, providing clear evidence of convergent evolution. Despite severe genomic constrains imposed by genome reduction and limited HGT, gene duplications, interface variations, domain fusions, and repeat expansions have helped the shaping effector fold and diversity. While the overall effector repertoire of phytoplasmas appeared largely unique, some PhAME domains share similarities with proteins from other mollicutes and pathogens. Collectively, our findings shed light on how genome-reduced bacteria innovate molecular functions and offer insights into phytoplasma biology, effector evolution, and host-pathogen dynamics. They also lay the groundwork for protein engineering approaches aimed at discovering or designing novel biomolecules with biotechnological potential.

**Data availability statement:** All data have been provided and uploaded as part of the main manuscript, in the supplementary data, or updated into a repository. AlphaFold predicted structures and mature PhAME sequences have been deposited in Zenodo (doi: https://doi.org/10.5281/zenodo.17379298). Further data are provided in the supplementary information (S1-S12 Tables).

**Funding:** This study was supported by the UK Research and Innovation (UKRI) Engineering and Physical Sciences Research Council EP/X024415/1 (awarded via the European Research Council (ERC) to SAH) (FGM, STM, SAH), Biotechnology and Biological Sciences Research Council (BBSRC) BB/P012574 (Plant Health ISP) (SAH, STM, VT), BBS/E/J/000PR9797 (Plant Health ISP – Susceptibility) (SAH, STM, VT) and BBS/E/JI/230001B (Advancing Plant Health ISP - Mitigating biotic threats to plant health) (FGM, STM, MM, SAH) with additional support from the John Innes Centre (JIC) Knowledge Exchange (KEC) Innovation Funds (VT), John Innes Foundation (FGM, STM, VT, MM, SAH) and the Gatsby Charitable Foundation (FGM, SAH). The funders had no role in the study design, data collection and analysis, decision to publish, or preparation of the manuscript.

**Competing interests:** We have read the journal's policy and the authors of this manuscript have the following competing interests: SAH receives funding from industry on phytoplasma effector biology. Two patents based on the SAP05-mediated ubiquitin-independent degradation have been published (International Publication Numbers: WO2022/129621 and WO2024/256685). FGM, STM, VT, and SAH have also filed patent applications related to the work described in this manuscript.

## Author Summary

Phytoplasmas are minute bacteria that inhabit plants and their sap-feeding insect vectors. Infected plants often show striking developmental abnormalities, such as excessive shoot proliferation and the conversion of flowers into leaf-like structures (phyllody). These changes frequently lead to sterility, redirecting the plant resources in ways that facilitate bacterial transmission. Because phytoplasmas depend on their hosts and rarely get an opportunity to pick up new virulence genes from other bacteria, their virulence factors likely diversify through other means, evolving features that let them fine-tune plant and insect-vector processes, including dampening their immune responses. To explore how such diversity has arisen, we analysed 239 phytoplasma genomes and identified a large repertoire of candidate virulence factors, which we term Phytoplasma Effectors (PhAMEs), encompassing established factors and many previously unrecognised ones. By predicting and studying their structures, we uncovered strategies of molecular innovation and gained insight into how these effectors interact with host proteins. Comparative analyses against proteins from other pathogenic bacteria, including those that infect humans, revealed both conserved and lineage-specific strategies of host manipulation. Our findings show how even the most genome-reduced bacteria can evolve sophisticated tools to manipulate their multicellular hosts, offering new perspectives for biology and biotechnology.

## Introduction

Obligate host-associated bacteria frequently undergo population bottlenecks and experienced extensive genome reduction [1,2]. Their restricted lifestyles also limit opportunities for horizontal gene transfer (HGT) between strains. Many well-characterized obligate symbionts have evolved toward mutualism. While they remain dependent on host-derived nutrients, they frequently provide essential metabolites, such as amino acids, to their hosts [3,4]. Others, such as *Wolbachia*, have become highly specialized in manipulating specific processes, including reproduction, immunity, and development, to promote their own transmission to the next generation of invertebrate hosts [5,6]. Intriguingly, some obligate bacteria with highly reduced genomes, such as phytoplasmas and other mollicutes, possess an arsenal of virulence factors, known as effectors, able to modulate multiple processes of their insect and divergent plant hosts in sophisticated ways [7–9]. This presents an evolutionary paradox: how do bacteria with minimal genomes, subject to repeated bottlenecks and limited opportunities for genetic exchange, maintain and evolve a diverse, adaptable effector repertoire to modulate host physiology?

Mollicutes exemplify extreme evolutionary streamlining. Descended from one or more Gram-positive *Clostridium*-like ancestors [10], they have lost their cell wall and most lack major metabolic pathways, resulting in cells with a single membrane that are entirely dependent on nutrients provided by their hosts for survival [11]. This

group includes phytoplasmas, a monophyletic lineage of insect-transmitted plant pathogens within the family Acholeplasmataceae. Together with acholeplasmas, phytoplasmas form an early-diverging clade within the Mollicutes [12]. A separate early-diverging lineage includes mycoplasmas, which are common pathogens of vertebrates, including humans, and spiroplasmas and entomoplasmas, which primarily colonize invertebrates and occasionally infect plants [13]. The deep divergence of the two clades is reflected in codon usage. Spiroplasmas, mycoplasmas, and their relatives have reassigned the UGA stop codon to encode tryptophan, an adaptation absent in acholeplasmas and phytoplasmas, which retained the canonical genetic code [14].

Mollicutes also include some of the smallest bacterial genomes capable of replication in axenic culture [15–17], though many, including phytoplasmas, have remained largely non-culturable. Due to their obligate host association, mollicutes experience frequent population bottlenecks that limit opportunities for HGT [18]. Nevertheless, they have evolved highly dynamic mobile genetic elements, some of which facilitate HGT across distantly related strains [19–22]. This genetic mobility challenges traditional views of genome reduction in obligate symbionts and pathogens and raises key questions about how mobile elements shape the evolution of virulence and symbiosis in mollicutes.

Phytoplasmas rely on both plant and insect hosts for dispersal [23,24] and are among the few bacteria that colonize the cytoplasm of plant cells, specifically targeting phloem sieve cells, while also invading and colonizing various tissues of their insect vectors [25]. Despite their highly compact genomes, ranging from 0.6 to 0.96 Mb and lacking many conserved metabolic genes [26–28], they carry a diverse set of effector genes that often lie within mobile genetic elements, called Potential Mobile Units (PMUs) [19,22,27]. These bacteria induce striking symptoms in their plant hosts, including shoot and leaf proliferations (known as witches' brooms symptoms), formation of leaf-like flowers (virescence and phyllody) and the retention of juvenile characteristics manifested as an extended leaf-producing vegetative developmental phase (neoteny) [29–31]. These symptoms often make the infected plants more attractive to the insect vectors promoting the dissemination of phytoplasma [31].

Phytoplasmas carry a distinctive repertoire of virulence factors and effectors that induce these symptoms. Among the functionally characterized effectors, a theme of hijacking host cellular processes has emerged, with phytoplasma effectors often bringing together host proteins regulating distinct processes. For example, the Aster Yellows strain Witches' Broom (AY-WB) phytoplasma (*Ca*. Phytoplasma asteris) effectors, secreted AY-WB proteins (SAPs) [32], includes the well-characterized SAP05, which acts as a molecular scaffold to bring together the 26S proteasome component RPN10 and the Zn-finger domains of SPLs and GATAs, leading to degradation of these transcription factors [30,33]. The action of SAP05 results in delayed plant aging and excessive leaf and shoot proliferations, symptoms that resemble witch's broom symptoms of phytoplasma-infected plants. SAP11 effectors induce stem proliferation characteristic of witches' broom symptoms by destabilizing TCP transcription factors, thereby promoting leaf shape changes and the outgrowths of additional stems [34–36] and impacting plant defence [36–38].

Another example is SAP54/phyllogens (Phyl1), which bind MADS-box transcription factors (MTFs) and the proteasome shuttle factors RAD23, resulting in the degradation of the former [39,40–43]. Since MTFs play a crucial role in flower development, this disruption transformed flowers into leaf-like structures, phenotypes that resemble phyllody and virescence, symptoms that are diagnostic for phytoplasma presence in plants. Remarkably, both SAP54 and SAP05 induce degradation in a ubiquitin-independent manner, bypassing conventional host regulatory mechanisms [30,41,43]. Phytoplasma SAP effectors not only reshape plant morphology but also enhance insect vector attraction and fecundity [41,44,45], ensuring efficient pathogen transmission.

Other functionally characterized phytoplasma effectors include TENGU [46], SWP12 [47], SRP1 [48], SJP39 [49] and RY378 [50]. Moreover, phytoplasma trans-membrane proteins such as AMP [51,52], IMP [53] P38 [54] and PM19_00185 [55] play key roles in phytoplasma colonization of their host.

While several phytoplasma effectors have been characterized, their overall diversity and structural features remain largely unknown. To investigate how phytoplasmas may diversify their effector repertoires under extreme genomic and

evolutionary constraints, we analysed the effectorome diversity of 239 phytoplasma genomes, using AlphaFold structural predictions to assess how effector domain composition, protein-protein interaction surfaces, and structural motifs may have evolved. We found that phytoplasma effectors diversify through variations on specific protein folds. For instance, the SAP05-fold appears in multiple effectors with differences in binding surfaces and repeat number. Small α-helical structures resembling those of SAP54/Phyl1 effectors are also common, consistent with their stability and binding versatility. We identified a new effector family, Scorpions, characterized by highly variable tandem β-hairpins, and discovered conserved folds shared across phytoplasmas, other mollicutes, and intracellular pathogens. These findings reveal how effector diversity evolves and offer insights relevant to protein engineering.

## Results

### Mining phytoplasma genome sequences identified 7,162 putative Phytoplasma Effectors (PhAMEs)

To explore the structural and sequence landscape of phytoplasma-secreted proteins, hereafter referred to as putative Phytoplasma Effectors (PhAMEs), we developed a dedicated bioinformatic pipeline. This pipeline enabled the integration of established approaches for phytoplasma effector identification [32] in 239 available phytoplasma genomes, followed by AlphaFold v2.0 (AF2)-based structural predictions of selected representative proteins [56] and the construction of a network of clustered PhAMEs based on both sequence and structural similarities. We split the phytoplasma genomes into two groups. One group of 24 representative phytoplasmas (RePh) represented the major phytoplasma 16Sr groups and were selected based on annotations of complete/chromosome level assemblies or high genome completeness at ≥ 90% CheckM completion [57] (S1 Table). The other group comprised 215 phytoplasmas and included fully assembled or partial genomes (S1 Table). Out of 107,543 full-length proteins encoded by any of the genomes, the pipeline identified 7,162 PhAMEs (2,205 distinct mature sequences) (6.7%) based on two criteria: the presence of a signal peptide involved in protein secretion by bacteria, including phytoplasmas [32]; and the absence of transmembrane domains in the mature protein (Fig 1A, S2 Table).

The number of PhAMEs, after removal of partial proteins (S3 Table), identified 5–67 proteins per phytoplasma (S1A Fig; S4 Table). Although genome assembly quality may partially account for variation in effector repertoire size, the presence of fully assembled genomes with both high and low effector counts indicated that this variation likely reflects true biological differences (S1B Fig).

### PhAMEs group into 556 clusters based on structure similarities

Functional annotation failed to identify functionally informative domains in most PhAMEs (S5 Table). Therefore, to establish homology relationships among the PhAMEs, we used MMseqs2 to group their protein sequences based on sequence identities and alignment coverages. This showed that the 7,162 PhAMEscan be divided into 894 sequence-based subclusters and singletons (sC), with the largest having 308 members (Fig 1A, S2A Fig, S4 Table). To enable structural analysis, 1,309 distinct representative PhAMEs were selected for prediction with AF2, ensuring coverage of all 894 sequence-based sub-clusters (Fig 1A). For sequence-based clustering and structure predictions, the mature protein sequences (without signal peptides) were used. Structural predictions quality was assessed using the average predicted local distance difference test (pLDDT) score as a confidence metric. This revealed that the majority of PhAMEs were predicted with high confidence, with 71.1% achieving pLDDT scores above 70 (S3A Fig, S4 Table). PhAMEs derived from RePh displayed a similar distribution of confidence scores between genomes, indicating that there is no obvious bias in the prediction quality of the secretomes. Of the 894 sub-clusters that were grouped based on sequence similarities, 598 (67.0%) included at least one member with a predicted structure having an average pLDDT score ≥ 70 (S5A Fig), indicating generally high-quality structural predictions across the phytoplasma secretome.

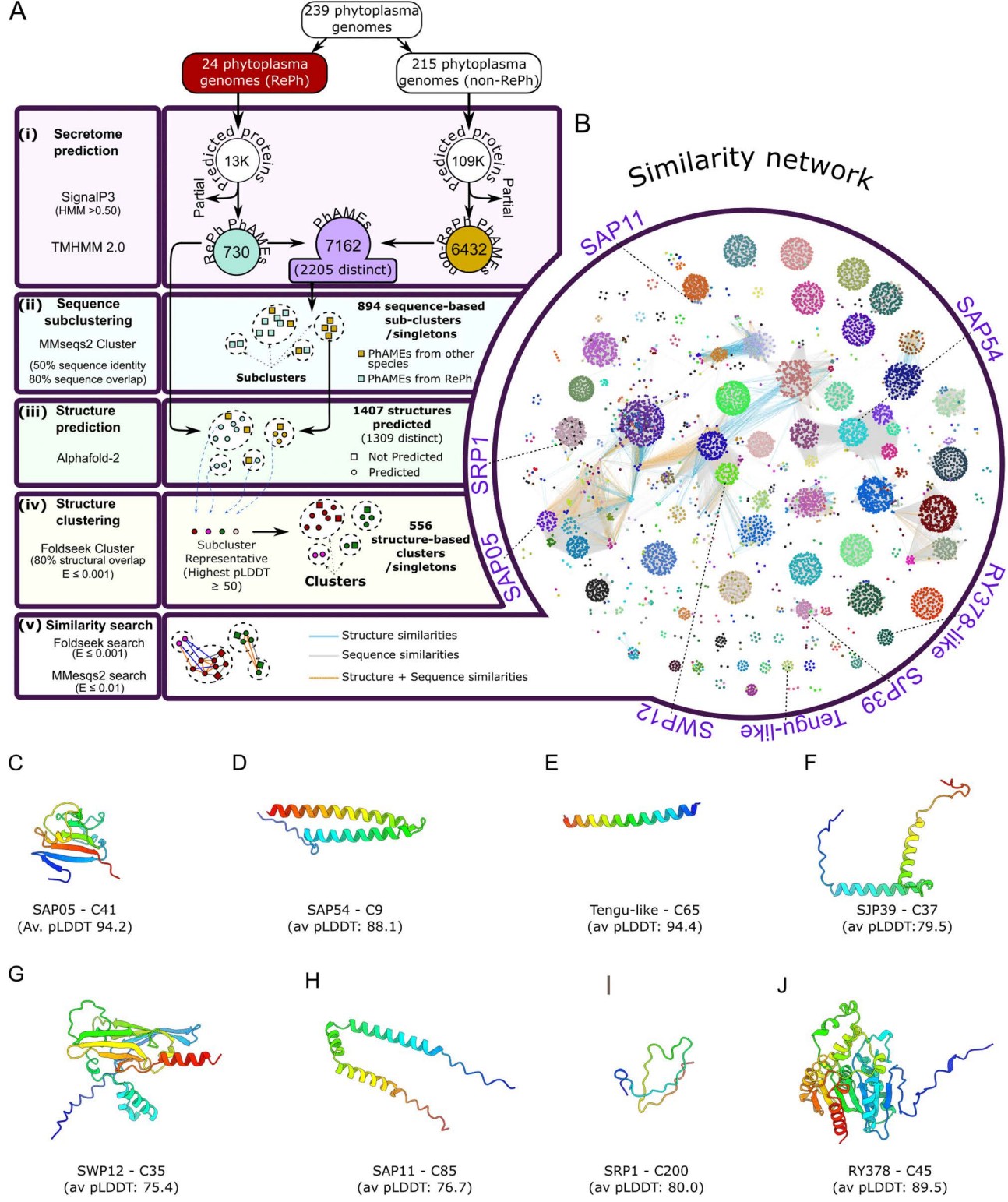

**Fig 1. PhAMEs group into clusters of structurally related proteins. (A)** Bioinformatic pipeline used in this study for the prediction, structural modelling, clustering, and similarity analysis of phytoplasma secreted proteins. **(i)** Secreted proteins were predicted from 24 representative phytoplasma (RePh) species and an additional 215 genomes using SignalP 3.0. Proteins with predicted signal peptides and no transmembrane domains were

classified as putative Phytoplasma Effectors (PhAMEs). Proteins annotated as partial were removed from the analysis. **(ii)** PhAMEs were clustered into sequence-based sub-clusters using MMseqs2, with thresholds of 80% sequence overlap and a minimum of 50% sequence identity. **(iii)** Structural predictions were performed for all PhAMEs encoded by RePh genomes, as well as for three additional proteins from sub-clusters not represented in RePh. For sub-clusters lacking representatives with an average pLDDT score ≥ 50, three additional structures were predicted. **(iv)** To generate structure-based clusters, sub-cluster representatives (defined as the protein with the highest average pLDDT score in each sub-cluster and average pLDDT score ≥ 50) were clustered using the Foldseek cluster algorithm (80% structural alignment, E-value ≤ 0.001). Each PhAME that belonged to the same sequence-based sub-cluster as a representative was assigned to that representative's structure-based cluster. **(v)** Structural similarities among all PhAMEs with an average pLDDT ≥ 50 were assessed using Foldseek (E-value ≤ 0.001), while sequence similarities were evaluated using MMseqs2 (E-value ≤ 0. 01). **(B)** Similarity network, showing structure and sequence relationships among the PhAMEs. Nodes represent individual PhAMEs, with node colour indicating cluster membership. Edges denote pairwise similarity: grey for sequence similarity only (MMseqs2, E-value ≤ 0.01), blue for structural similarity only (Foldseek, E-value ≤ 0.001), and orange for both sequence and structural similarity. Unconnected nodes represent proteins with no detectable sequence or structural similarity to any other PhAME. Unconnected singletons are not represented. Experimentally validated effectors or effector families are labelled with long dashed lines and named. The genome of *Ca*. Phytoplasma oryzae strain HN2022 was sequenced and characterized during the preparation of the manuscript and thus RY378 was not included in the similarity network, but cluster membership of RY378 was evaluated retrospectively. The location of the RY378-like WBL31615.1 from SCWL1 (86.7% sequence identity, 99.3% query coverage to RY378) is indicated instead. Only protein pairs with predicted structures for both proteins were included in the structural similarity analysis. **(C-J)** AlphaFold-predicted structural models of experimentally validated phytoplasma effectors coloured from blue at the N-terminal end to red at the C terminus. Cluster membership and average pLDDT scores are indicated.

To further verify the accuracy of our structure predictions, we performed structure-based alignments between available experimentally determined effector structures and their corresponding AF2 models. Crystal structures have been obtained of two secreted phytoplasma effectors and their homologs: SAP05 exhibits a characteristic globular structure with a β-sheet core [33,58,59] and SAP54/Phyl1 has an antiparallel coiled-coil fold [42,60,61]. Structural alignment between experimentally determined crystal structures and their corresponding AF2 models revealed high similarity, with RMSD values of 0.74 Å for SAP05[AY-WB] and 0.70 Å for SAP54[OY-M]/Phyl1 (S4 Fig), indicating strong concordance between predicted and experimental structures.

Sequence-based clustering risks the failure to identify relationships between PhAMEs with low sequence identities that nonetheless share similar folds and domain architectures. To address this, we also performed structure-based clustering of the representative PhAME structural models. For each sequence-based sub-cluster (sC) containing members with average pLDDT scores > 50, the highest-ranking model was selected as its structural representative. These representatives were then clustered using Foldseek with an all versus all search [62], to group the PhAMEs into structure-based clusters. The two clustering approaches were combined by assigning all PhAMEs within each sequence-based sub-cluster to the same structure-based cluster as their respective representative (Fig 1A), as proteins with 50% sequence identity are very likely to have the same fold [63]. This integrative clustering approach reduced the original 894 sequence-based sub-clusters/singletons into a more compact set of structure-based clusters. Of the 7,162 identified PhAMEs, 6,840 (95.5%) were assigned to one of 556 structure-based clusters or designated as structural singletons (Fig 1A, S4 Table). Notably, 71.4% of these clusters and singletons included at least one PhAME with an average pLDDT score ≥ 70 (S5B Fig). Structure-based clusters encompassed between 1 and 34 distinct sequence-based subclusters (S5C Fig), highlighting cases of strong structural similarities despite low sequence conservations. Furthermore, 80.5% of the clusters containing four or more members contained PhAMEs from at least one RePh, illustrating that the 24 RePh phytoplasmas collectively captured most of the structural diversity present in the broader pan-effectorome (S5D Fig).

## The PhAME structural clusters include functionally characterized effectors

We next wondered if the PhAME clusters included phytoplasma effectors for which functions have been described (Fig 1B–J). Of these, SRP1 [48] was found to be a singleton, while SAP11 [34,35], SAP54 [39], SAP05 [30], TENGU [46], SJP39 [49] and SWP12 [47] were all found to belong to broader clusters that include structurally and sequence-related PhAMEs (Fig 1B, S4 Table) underscoring the robustness of our approach. However, some homologues of previously identified effectors were not identified by our pipeline and so were not present in our set of PhAMEs. These included,

for example, the SAP05 homologues from Witches-Broom Disease of Lime (WBDL; *Ca.* Phytoplasma aurantifolia) phytoplasma. Upon closer inspection, we noticed that these could be grouped into three categories: i) incorrect start codon predictions; ii) lack of an obvious start codon; or iii) lack of predicted secretion signals, likely due to false negatives from existing secretion prediction tools [64]. Nonetheless, given that our PhAME network included all functionally characterised phytoplasma effectors, our approach appeared to have captured the vast diversity of the phytoplasma effector repertoire.

During the preparation of this manuscript, RY378 from *Ca.* Phytoplasma oryzae was shown to induce rice tillering and predicted to adopt an α/β-hydrolase fold [50]. Our analysis supported this and links RY378 to PhAME cluster C45 (Fig 1B, 1J). Proteins with this fold, such as *M. tuberculosis* Rv0183 (PDB 6EIC), are known virulence factors involved in host lipid metabolism [65]. Whether RY378 possesses enzymatic activity or operates via a non-catalytic mechanism remains to be determined.

## The PhAME structural network reveals interconnected clusters with shared domains

Next, we assessed the PhAMEs with shared common domains, including an all-against-all structural comparison using the Foldseek [62], complemented by sequence similarity searches using MMseqs2 [66]. This required removal of the threshold percentage of overlap between the sequences so that protein pairs sharing a single domain (representing only a portion of one or both proteins) might be identified (Fig 1A). Notably, several clusters exhibited high interconnectivities (Fig 1B). For example, SAP05 effectors, classified within cluster C41, formed part of a connected network of clusters that shared a SAP05-like fold fused to additional domains. One such example is SAP49, a member of cluster C28 (S6A Fig), which we found to contain a SAP05-like fold with a C-terminal α-helix. Similarly, the SAP54 cluster (C9) displayed structural similarity to members of other clusters characterized by α-helical folds, such as those in cluster C48 (S6B Fig). In addition, we identified a large group of interconnected clusters with no functionally characterized members to date. Proteins within these sub-networks share a distinctive β-sheet-rich domain architecture, typically accompanied by N-terminal α-helical extensions (S6C Fig), such as PhAMEs with membership to C3 and C46. Thus, our pipeline enabled the structural profiling of the phytoplasma pan-effectorome, grouping effectors based on both sequence and structural similarities. Given this, we proceeded to use the dataset to better understand the evolution of functional diversity amongst phytoplasma effectors.

## SAP05 effectors group together in the C41 structural cluster

We identified cluster C41 as comprising the SAP05 family of effectors, including both previously characterized members [30] and several newly predicted homologs (S7A-B Fig). SAP05[AY-WB] has a globular compact structure with five β-strands that form an internal triangular mixed β-sheet core [33]. The surface dominated by three loops (L4, L5 and L7) binds GATA and SPL transcription factors, whereas the surface on the opposite site of the SAP05 protein composed of ß-sheets 1 and 5, α-helices 1 and 2, and L2 and 3 interacts with the vWA domain of the proteasomal subunit RPN10 (Fig 2A; [33]). The AF2-predicted protein structures of all C41 members closely aligned that of SAP05[AY-WB] in agreement with these proteins also binding GATA and SPL transcription factors and RPN10 [30,33] (S7C Fig). The loop surface exhibits greater sequence variability compared to structures composing the protein core (S7D Fig), in agreement with previous findings that some SAP05 homologs have specificity to only bind SPLs or GATAs, including for instance SAP05 of *Ca.* Phytoplasma mali [30,33]. These data show that our AF2-based structure predictions and clustering methods match experimental data, thereby validating our pipeline and approach to assess the structural diversity of phytoplasma effectors.

## The loop surface structure of SAP05 mimics the major DNA groove recognized by the DNA-binding interface of GATA transcription factors

Zinc-finger domains of GATA and SPL transcription factors mediate DNA binding [68]. Given that SAP05 also interacts with Zinc-finger domains [30], we compared the structural features of GATA zinc fingers bound to SAP05 (S7E Fig) [58] with those bound to DNA [69] (S7F Fig). These comparisons revealed that GATA transcription factors use the same

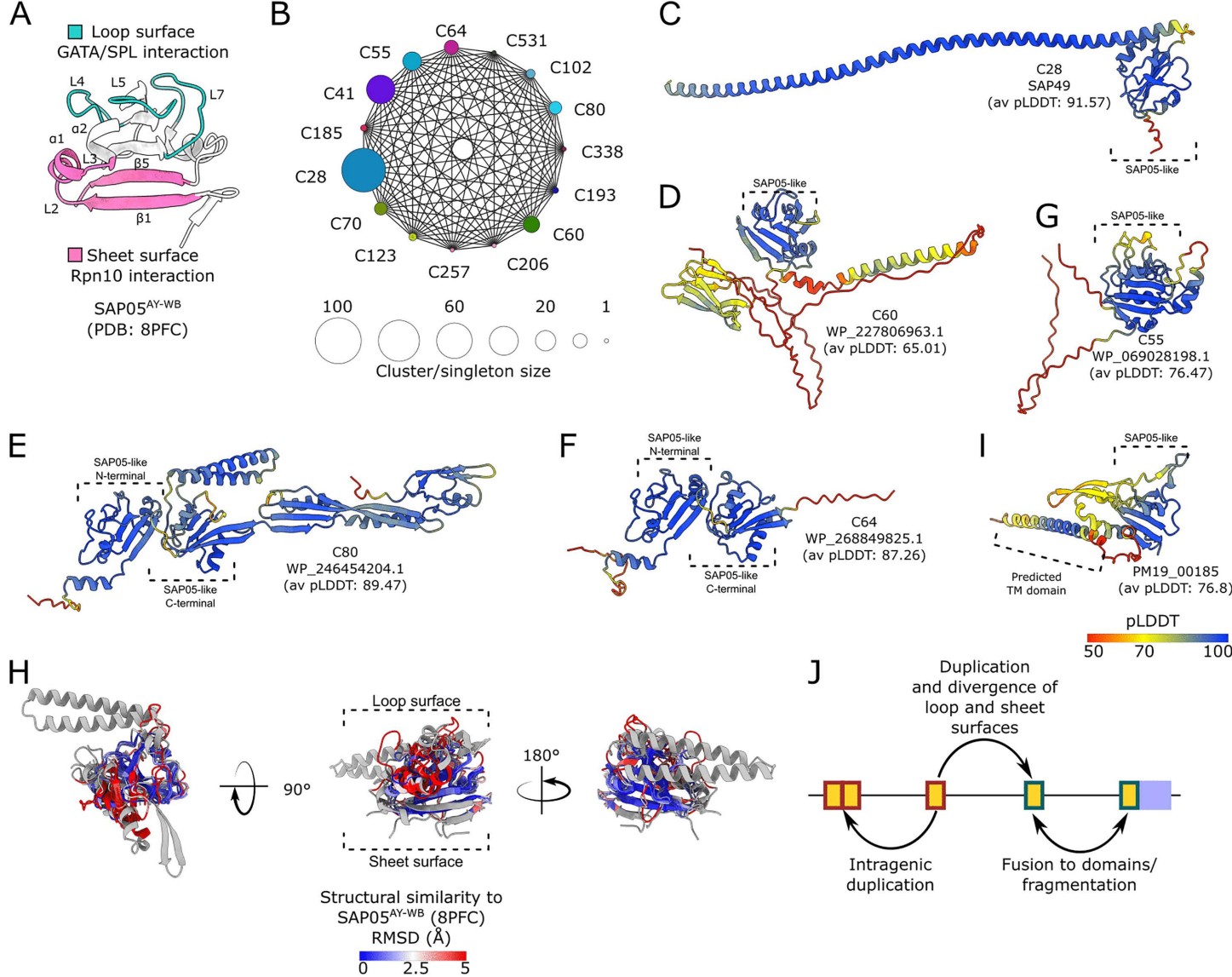

**Fig 2. SAP05-like proteins exhibit conserved core folds with divergent surface-exposed regions. (A)** Crystal structure of SAP05^AY-WB effector [33]. The loop region involved in interaction with transcription factors (TFs) is highlighted in cyan, and the β-sheet-rich region involved in interaction with RPN10 is shown in pink. Secondary structure elements participating in these interactions are indicated. **(B)** Structure similarity network of proteins with SAP05-like folds. Structure-based cluster numbers (C28, C41, etc) are indicated next to the nodes, with node size representing the number of PhAMEs within each cluster or singleton. Edges denote structural similarity between cluster members, as determined by Foldseek (E-value ≤ 0.001). **(C-G)** AlphaFold v2.0 (AF2)-predicted structures of select proteins with SAP05-like folds, coloured by per-residue pLDDT value. The presence of one or more SAP05-like domains within each of the proteins is indicated. The text below each structure indicates cluster number (top), protein name (middle) and average AF2 pLDDT scores (bottom). **(H)** Superimposition of the predicted structures of SAP05-like domains and their level of similarities to the crystal structure of SAP05^AY-WB, coloured by root mean square deviation (RMSD), using the Matchmaker command from ChimeraX, revealing a conserved core structure with variable loop regions. Gray indicates RMSD > 5. For each structure-based cluster and singleton, the model with the highest average pLDDT score was used for superimposition. SAP05-like domains were defined using sequence-independent multiple structure alignment with FoldMason [67], additional residues were included after the alignment to capture the full extent of the β5 β-sheet. **(I)** Predicted structure of the membrane-bound E3 ubiquitin ligase PM19_00185 of *Ca.* Phytoplasma mali, coloured by pLDDT value. **(J)** Schematic overview of evolutionary processes that led to the amplification and diversification of proteins with SAP05-like folds.

surface to interact with both DNA and SAP05. Notably, we observed that the SAP05 loop surface structurally resembles the DNA major groove surface recognized by GATA transcription factors (S7G-H Fig). Moreover, key GATA residues involved in DNA base recognition also mediate SAP05 binding (S7H Fig) [69]. Thus, the SAP05 loop surface appears to mimic the major groove of DNA targeted by GATA transcription factors.

## The SAP05-like fold serves as a diversification platform for PhAMEs

A total of 14 clusters were identified that share sequence and structural similarities to SAP05 members of cluster C41 (Fig 2B). These clusters include proteins with domains exhibiting SAP05-like folds (Fig 2C–F). Although these SAP05-like domains show high structural similarities, they have low sequence conservation (S8A Fig), which likely explains why they were not previously recognized as SAP05 homologs. Within C41, MDV3198312.1 is the sole representative of the sequence-based sub-cluster sC535 and has a distinct sequence in the L4 region (S7B Fig), a characteristic that is also reflected in the AF2-mediated structure prediction (S7C Fig). Given that L4 of SAP05 has a crucial role in binding the Zinc-finger domains of both SPLs and GATAs [33], MDV3198312.1 is unlikely to bind these TFs.

Among related clusters to C41, C28 was the largest, containing 102 members. It includes PhAMEs featuring a SAP05-like domain fused to a C-terminal α-helix, exemplified by SAP49 from the AY-WB phytoplasma (S8C Fig). SAP05-like domains in these clusters exhibit various architectures: they are fused to structured domains of unknown function (S8D-E Fig); arranged as tandem repeats (S8E–F Fig); or linked to likely intrinsically disordered regions (S8G Fig). Comparisons of the SAP05$^{AY-WB}$ structure with the predicted structures of SAP05-like domains in other clusters shows high conservation of the β-sheet internal core, but highly divergent structures at surface areas, particularly in the SPL/GATA- and vWA-binding surfaces of SAP05 (Fig 2H, S8 B-F Fig). Altogether, these findings indicate that the SAP05-like fold serves as a diversification platform for PhAMEs by varying interaction interfaces while preserving a conserved structural core. Furthermore, genes encoding SAP05-like folds have expanded in phytoplasma genomes and frequently acquired additional domains or intrinsically disordered regions (Fig 2J), potentially facilitating interactions with a wider array of host targets.

To identify additional SAP05-like fold-containing proteins across phytoplasmas, we performed a structure-based search using Foldseek against AlphaFold-predicted models [70] from *Ca.* Phytoplasma species (NCBI Taxonomy ID: 33926), comprising 16,731 predicted structures, including complete proteomes from several species (S6 Table). The search revealed numerous proteins with SAP05-like folds, many of which lack obvious signal peptides (S6 Table). One of these is the membrane-associated protein PM19_00185 from *Ca.* Phytoplasma mali (Fig 2I) that interacts with several plant ubiquitin-conjugating enzymes (UBC family) and has been proposed to exhibit E3 ubiquitin ligase activity [55]. Stable expression of PM19_00185 in plants increases susceptibility to the non-host pathogen *Pseudomonas syringae* pv. *tabaci* [55], supporting its role as a virulence factor. ATP_00103 from *Ca.* Phytoplasma mali is another protein containing a SAP05-like domain and is annotated as a putative telomere resolvase [28]. It is intriguing that this protein is present in *Ca.* Phytoplasma mali, whose genome is unusual among phytoplasmas in consisting of a linear chromosome flanked by telomere-like sequences, in contrast to the typically circular genomes of most other phytoplasma species. Overall, this indicates that the SAP05-like fold is highly versatile, and its diversification allows for new virulence and other functions.

## SAP54/Phyl1 PhAMEs group together in the C9 structural cluster

SAP54/Phyl1-like proteins promote the degradation of plant MTFs by simultaneously interacting with the K-domain of the transcription factors and the ubiquitin-associated (UBA) domain of the proteasomal shuttle protein RAD23. This dual interaction enables the recruitment of RAD23, which facilitates the delivery of the transcription factor cargo to the 26S proteasome for degradation [41]. Our pipeline identified 169 SAP54/Phyl1-like proteins that cluster within group C9 and include both characterized and uncharacterized members. Phylogenetic analysis of proteins within the C9 group reveals five clades supported by high bootstrap values, consistent with previous reports [71] (S9A Fig). The predicted structures of proteins in the C9 group closely resemble those of SAP54/Phyl1 from the OY-M and PnWB phytoplasmas (S4C-D,

S9B Fig) [42,60]. These data provided further confidence in our pipeline and approach to assess the structural diversity of phytoplasma effectors.

## Opposite facing surfaces of SAP54/Phyl1 interact with MADS-box transcription factors and the proteasomal shuttle protein RAD23

The interaction of SAP54 with MTFs and RAD23 has been shown experimentally [40–42] However, the structural bases of the interacting complex have not been resolved. We modelled the structure of SAP54/Phyl1 from 'Ca. Phytoplasma asteris' OY-M in complex with A. thaliana SEP3 (AtSEP3) and AtRAD23C using AF-M (AlphaFold-Multimer V.2). SAP54^OY-M/Phyl1 was selected because the structure has been experimentally determined [42]. The model correctly identified the AtSEP3 K-domain as the interaction domain in the AtSEP3-SAP54^OY-M/Phyl1 complex, as has previously been experimentally shown [41,43]. Specifically, a surface involving the K-domain residues 146–175 (S10A Fig) were found to interact with SAP54^OY-M/Phyl1 at high confidence (ipTM = 0.845), in agreement with experimental data [43]. The model shows that SAP54^OY-M/Phyl1 interacts with helix 2 of the SEP3 K-domain via a composite surface formed by its two α-helices, involving charged and uncharged residues (Figs 3A, S10A–C). Structural comparison with the multimeric SEP3 crystal structure (PDB: 4OX0) revealed that SAP54^OY-M/Phyl1 binds the K domain at the same interface used in SEP3 multimerization (S10D–F Fig). This finding reinforces earlier results showing that SAP54 disrupts the multimerization of MTFs [60], and that mutations in SEP3 residues essential for tetramerization prevent Phyl1 binding [43].

The AF model also confidently predicts that SAP54^OY-M/Phyl1 interacts with the C-terminal UBA (UBA2) domain of AtRAD23C (ipTM = 0.854 for full length RAD23; ipTM = 0.903 for C-terminal UBA domain) (S10G-I Fig), consistent with data herein (Fig 3B) and previous reports [43]. Structural evaluation of the model revealed that the negatively charged surface of the RAD23C C-terminal UBA domain (Fig 3C) interacts with the positively charged surface of SAP54^OY-M/Phyl1.

Given that the RAD23 UBA domains are known to bind ubiquitin [72], we superimposed the predicted SAP54^OY-M/Phyl1 model and ubiquitin bound to the autophagy receptor NBR1 (neighbour of BRCA1 gene 1) UBA domain, for which there is a crystal structure (PDB: 2MJ5; S10J Fig). This revealed that Phyl1/SAP54^OY-M binding to UBA2 likely overlaps with that of ubiquitin (S10K Fig).

The UBA2- SAP54^OY-M/Phyl1-AtRAD23C structure model shows that opposing surfaces of SAP54^OY-M/Phyl1 interact with the UBA domain of AtRAD23C and the K domain of AtSEP3 (Fig 3D), suggesting that the effector oppositely oriented interfaces enable it to shuttle MTFs to the proteasome by recruiting RAD23. This mechanism is reminiscent of SAP05-mediated degradation of SPL and GATA transcription factors via recruitment of the 26S proteasome component RPN10 [33,58], although SAP05 binds the vWA domain and not the ubiquitin-interacting domain of RPN10.

## Structural model validation reveals conserved SAP54/Phyl1 residues mediating MADS-box and RAD23 binding

To validate the structural models, we introduced targeted mutations into SAP54^AY-WB and analysed its interactions with RAD23 proteins and the MTF SEP3 using yeast two-hybrid assays (S7 Table). SAP54^AY-WB and SAP54^OY-M share 84% amino acid sequence identity and exhibit a high degree of structural similarity (RMSD = 0.54 Å over 67 pruned atom pairs).

SAP54^AY-WB K61E (helix A), K106E and K110 (helix B) mutants did not interact with RAD23C, whereas their binding to the MTFs was not affected (Figs 3E, S11A).

Conversely, the N59Y mutation, predicted to disrupt the SAP54–SEP3 interface, abolished binding to MTFs while preserving interactions with RAD23C (Fig 3E-F). Consistent with these results, two previously identified SAP54^OY-M/Phyl1 residues involved in MTF binding [71] map to the same surface as N59 (Fig 3E-F). The N59Y mutation had a less profound effect on the binding of SOC1 than SEP3 and AP1, suggesting that the SAP54 residues required for interaction may vary depending on the specific TF target, in agreement with findings that SAP54/Phyl1 homologues have different binding specificities for MTF [71]. To investigate this further, we generated a multiple sequence alignment of C9 SAP54/

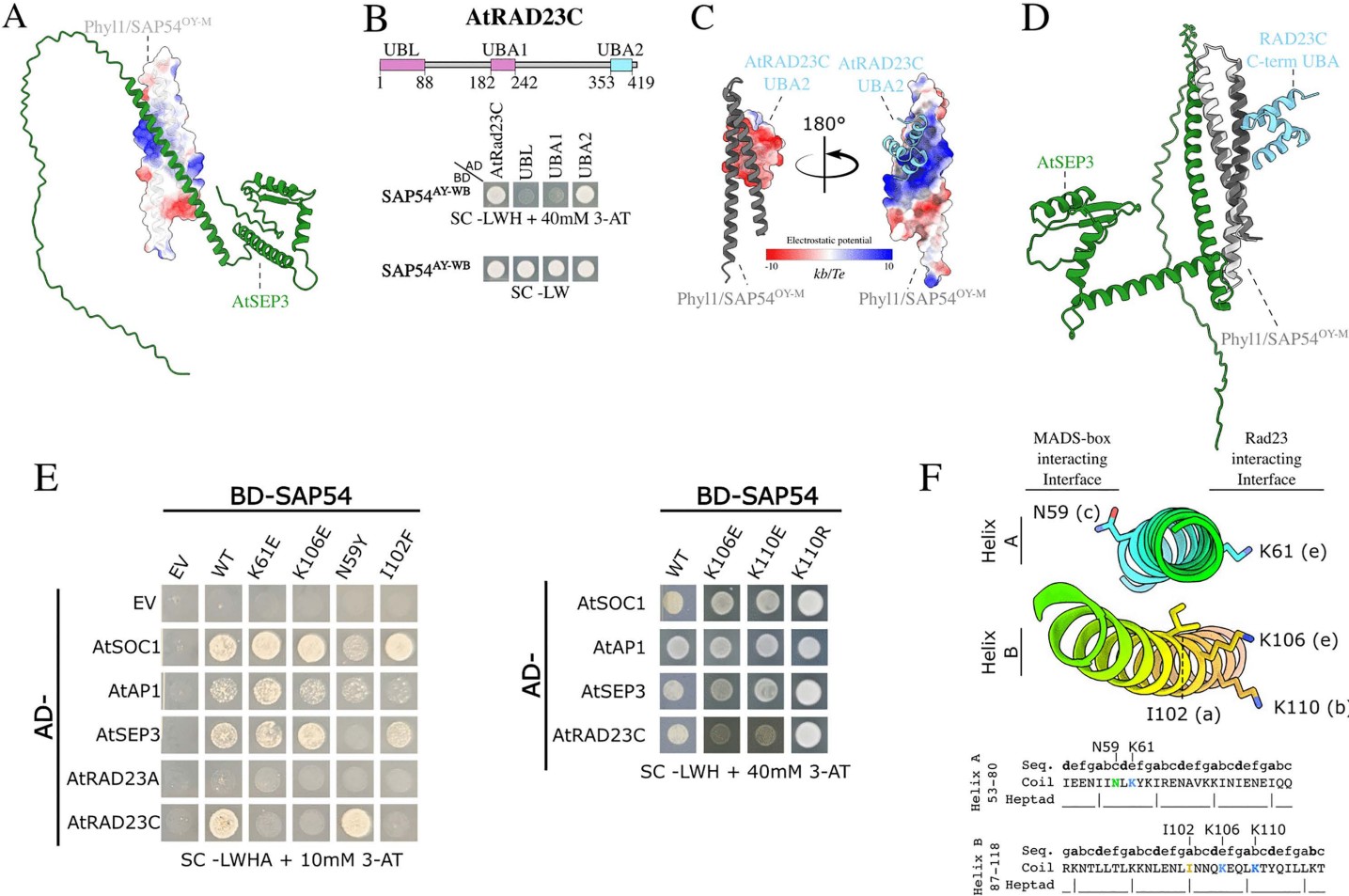

**Fig 3. Opposing surfaces of SAP54 mediate selective binding to the K-domain of MADS-box transcription factors and to the UBA domains of the proteasomal shuttle protein RAD23. (A)** Surface representations of the predicted SAP54[OY-M]/Phyl1–AtSEP3 complex, with Coulombic electrostatic potential mapped onto SAP54[OY-M]/Phyl1. Surfaces are coloured from positive (blue) to negative (red) charge. Electrostatic potential was calculated using ChimeraX. **(B)** Top: Schematic overview of the locations of the ubiquitin-like (UBL) and ubiquitin-associated (UBA) domains in *A. thaliana* RAD23C. Bottom: Yeast two-hybrid (Y2H) assay showing that SAP54[AY-WB] interacts with the UBA2 domain of RAD23C with its corresponding transformation controls. AD, GAL4 activation domain; BD, GAL4 DNA-binding domain; SD-LWH, selective medium lacking leucine **(L)**, tryptophan **(W)**, and histidine **(H)**; 3-AT, 3-amino-1,2,4-triazole, a competitive inhibitor of the HIS3 enzyme. EV, empty vector control. Growth on double dropout (SC-LW) medium confirms the presence of both constructs. **(C)** Surface representations of the predicted SAP54[OY-M]/Phyl1–UBA2(AtRAD23C) complex, with electrostatic potential mapped onto UBA2(AtRAD23C) (left) and SAP54[OY-M]/Phyl1 (right). Surfaces coloured as in **(A)**. **(D)** Predicted model of the AtSEP3–SAP54[OY-M]/Phyl1–UBA2(RAD23C) ternary structure with the K-domain of SEP3 and the UBA2 domain of RAD23C binding to opposing surfaces of SAP54. Model was obtained by superimposing the AlphaFold-Multimer predicted complexes of AtSEP3–SAP54[OY-M]/Phyl1 and SAP54[OY-M]/Phyl1–UBA2(RAD23C). **(E)** Left: Y2H assay showing that SAP54[AY-WB] K61E, K106E, and I102F mutants fail to interact with *A. thaliana* RAD23C, while SAP54[AY-WB] N59Y has reduced interaction with MTF. Right: Y2H assay showing that the SAP54[AY-WB] K110E mutant fails to interact with *A. thaliana* RAD23C, while SAP54[AY-WB] K110R, which maintains the positive charge of the residue, still interacts with the proteosomal shuttle protein. Numbering based on full-length/mature protein. SD-LWHA, selective medium lacking leucine **(L)**, tryptophan **(W)**, histidine (H) and Adenine **(A)**. **(F)** Locations of N59, K61, K106, K110 and I102 in the two helices A and B of the structure (top) and linear sequence (bottom) of SAP54[AY-WB]. The seven-residue sequence repeats (heptads) of the SAP54 helices are shown and assigned based on Iwabuchi et al., 2019 [42]. I102 at position a of the heptad locates in the centre of the helices, whereas N59, K61, K106 and K110 at positions c, e, e and b, respectively, are exposed towards the exterior of the helices. SAP54 is coloured from blue at the N-terminal end to red at the C terminus.

Phyl1 homologues (S12A–B Fig). Residues K61, K106 and K110 involved in binding the RAD23C UBA domain are highly conserved among the SAP54/Phyl1 homologues (S12A–B Fig). In contrast, residue N59, located on the MTF interaction surface, was found to be polymorphic (S12A–B Fig). Notably, *Ca*. Phytoplasma citri has two SAP54/Phyl1 homologs with one carrying the asparagine (N) residue and the other (WEX20375.1) a tyrosine (Y) at this position (S12A Fig), suggesting that some phytoplasma strains have more SAP54 homologs with different binding specificities for MTFs. Furthermore, in certain clade C members, this residue is replaced by a negatively charged amino acid, highlighting further diversification.

Additionally, the I102F mutation in SAP54^AY-WB abolished RAD23 binding and disrupted the interaction with AP1 and SEP3 consistent with the predicted role of I102 in stabilizing the helix A–B interface that contributes to the essential positively charged surface (Fig 3E-F). This highlights the critical importance of structural integrity for effector function, in agreement with previous findings [60].

## Proteins with SAP54-like folds contribute to phytoplasma effector diversity

We identified clusters that are structurally or sequence-related to SAP54 and other members of the C9 group in the phytoplasma effector structure similarity network (S13A Fig). Structural inspection showed that these clusters are composed of proteins with predominantly α-helical architectures, including predicted folds consisting of one to four helices (S13B-G Fig). Members of cluster C48, for example, are predicted to adopt a structure comprising two antiparallel coiled coils connected by a linker, resembling the SAP54 dimer configuration (S13G Fig). Consistently, sequence-based analysis indicates that these proteins arose through intragenic tandem partial duplication. Notably, we observed a mutual exclusivity in the distribution of C9 and C48 proteins across phytoplasma genomes, with those containing C9 members lacking C48 members and vice versa, suggesting that these proteins are functionally redundant or occupy similar roles (S4 Table).

To test the functional relevance of C48 members, we examined whether SAP54 from SPLL (MDV3139961.1), a representative of C48, interacts with RAD23 proteins. In yeast two-hybrid assays, this protein bound specifically to RAD23C and RAD23D, but not to RAD23A or RAD23B, mirroring the binding profiles of other SAP54 homologues (S13H Fig). These findings support the role of C48 members as functional effectors and highlight how tandem duplication contributes to effector diversification and structural innovation within PhAMEs characterized by compact α-helical folds (S13I Fig).

## Widespread α-helical architecture among PhAMEs

To further investigate the diversity of α-helical PhAMEs, we manually examined AF-predicted structures and identified several clusters composed of proteins with simple α-helical architectures. Notably, 49.8% of PhAME clusters are predicted to predominantly consist of α-helical folds (S14A–B Fig, S8 Table), exhibiting a variety of structural arrangements. This prevalence highlights both the functional importance and structural versatility of α-helical proteins in the phytoplasma effector repertoire. Given that such proteins often function as molecular binders rather than enzymes, this suggests that protein–protein interaction mediators constitute a major component of the phytoplasma effectorome.

Interestingly, PhAMEs from distinct clusters often adopt antiparallel coiled-coil folds. These include SAP68, SAP67, SAP66, and SAP56, effectors that are transcriptionally upregulated during plant infection [39] and co-expressed in a polycistronic unit with SAP11 [32]. Like SAP54, these proteins are predicted to form amphipathic helices with inward-facing hydrophobic residues (S14C Fig). However, they differ in surface charge distribution (S14D Fig), suggesting divergence in binding specificity. Consistently, it was previously shown that SAP68, SAP67, and SAP66 interact with distinct plant transcription factors in yeast two-hybrid assays compared to SAP54 [73], supporting functional divergence. These findings suggest that phytoplasmas exploit the antiparallel coiled-coil motif as a versatile binding scaffold, enabling interaction with diverse host targets through opposing surfaces, resembling attributes of SAP54.

Other effectors such as SAP11, TENGU-like, and SJP39 are also predicted to be small α-helical proteins but adopt distinct conformations, indicating different modes of host interaction (Fig 1E, 1F, 1H). SAP11, for example, is predicted

to contain three α-helices connected by two loops (S15A Fig) and binds TCP TFs via their helix-loop-helix dimerization domains [34–36,73]. This interaction destabilizes TCPs, promotes insect vector reproduction, and triggers developmental changes such as witches' brooms and altered leaf morphologies [36].

## Dimerization mimicry by SAP11-like proteins enables TCP binding

To further investigate the structural basis of SAP11 interactions with TCP TFs, we modelled the SAP11[AY-WB]–AtTCP10 complex using AlphaFold-Multimer. The model yielded an intermediate ipTM score (0.705), consistent with experimental evidence showing SAP11 binds the TCP dimerization domain (S15B Fig). In the model, SAP11[AY-WB] helices were found to engage with the helix-loop-helix region of AtTCP10, with the loops enabling the positioning of the helices. The predicted SAP11–TCP interface exposes a hydrophobic patch (S15C Fig) that resembles the dimerization surface of TCP proteins (S15D–E Fig; PDB: 7VP1; [74]). Structural comparison of the SAP11–AtTCP10 model with the AtTCP10 homodimer (PDB: 7VP1) further indicated that SAP11 may interfere with TCP homodimer formation by mimicking this hydrophobic surface (S15F Fig). This supports a model in which SAP11 exploits conserved dimerization interfaces to interact broadly with members of the TCP TF family.

In the effector network, SAP11-like proteins are distributed across clusters C8, C85, and C445, comprising 180, 6 and 1 members, respectively (S16A Fig, S4 Table), matching the previously identified SAP11 clades with distinct binding specificities to TB1, CIN and CYC TCP sub-family members [73]. Phylogenetic analysis with additional SAP11 homologs recovered these clades and additionally revealed three additional clades with high bootstrap support (88, 98, and 100; S16B Fig). Structural modelling showed that SAP11-like proteins consistently adopt an architecture comprising three-helices and two loops (S16C Fig), and multiple sequence alignments revealed conserved hydrophobic residues at the predicted TCP-binding interfaces among the SAP11 homologs (S16B, S16D Fig). These features support the model that SAP11-like proteins function as dimerization mimics, allowing them to interact with a broad, yet specific, subset of TCP TFs.

## 'Scorpion' PhAMEs diversify by sequence composition and number of β-hairpin repeats

Beyond PhAMEs that cluster into networks with SAP05-like and α-helical domains, we also uncovered ones with new structures. These include interconnected clusters of PhAMEs characterized by a distinct β-sheet architecture, often with N-terminal α-helical tails (Fig 4A, S6C Fig) that we named 'Scorpions'. Among these, C3 and C46 are the two clusters with most members, containing 284 and 22 PhAMEs, respectively (Fig 4A, S4 Table). Functional annotation revealed that many Scorpions contain partially overlapping, uncharacterized DUF2963 domains, which map to the β-sheet domain (S5 Table). Structural analysis indicated that this domain consists of multiple β-hairpin repeats. To better characterize them, we employed structure- and sequence-based tandem repeat prediction tools [76,77] followed by additional analyses (see materials and methods). We found that each repeat is about 26 residues, with certain positions showing high sequence conservations (Fig 4B). We named the repeats Scorpion β-hairpin (hp) motifs. Profile searches using HMMER v3.4 against the identified PhAMEs showed severalwith varying numbers of Scorpion β-hp motif repeats, ranging from 1 to 24 (Fig 4C, S9 Table). All PhAMEs with predicted DUF2963 domains were found to have Scorpion β-hairpin motifs, in agreement with their groupings in a distinct structural category in our AF2 pipeline. Notably, other PhAMEs with β-hairpins, including those with membership to clusters C3 and C46, were not found to have Scorpion β-hp repeats (Fig 4I-J), suggesting higher sequence divergence or structural convergence. Furthermore, certain phytoplasma species, such as AY-WB phytoplasma, encode six Scorpion PhAMEs with varying repeat numbers (Fig 4D-F), suggesting functional diversification. Interestingly, three of these proteins are encoded in plasmids and the other three in chromosomal regions near plasmid-like elements [27], suggesting that these may have derived from the integration of (partial) plasmids into the chromosome. Overall, we found a novel group of PhAMEs that have evolved to vary in repeat number and sequence, contributing to effector diversification (Fig 4G).

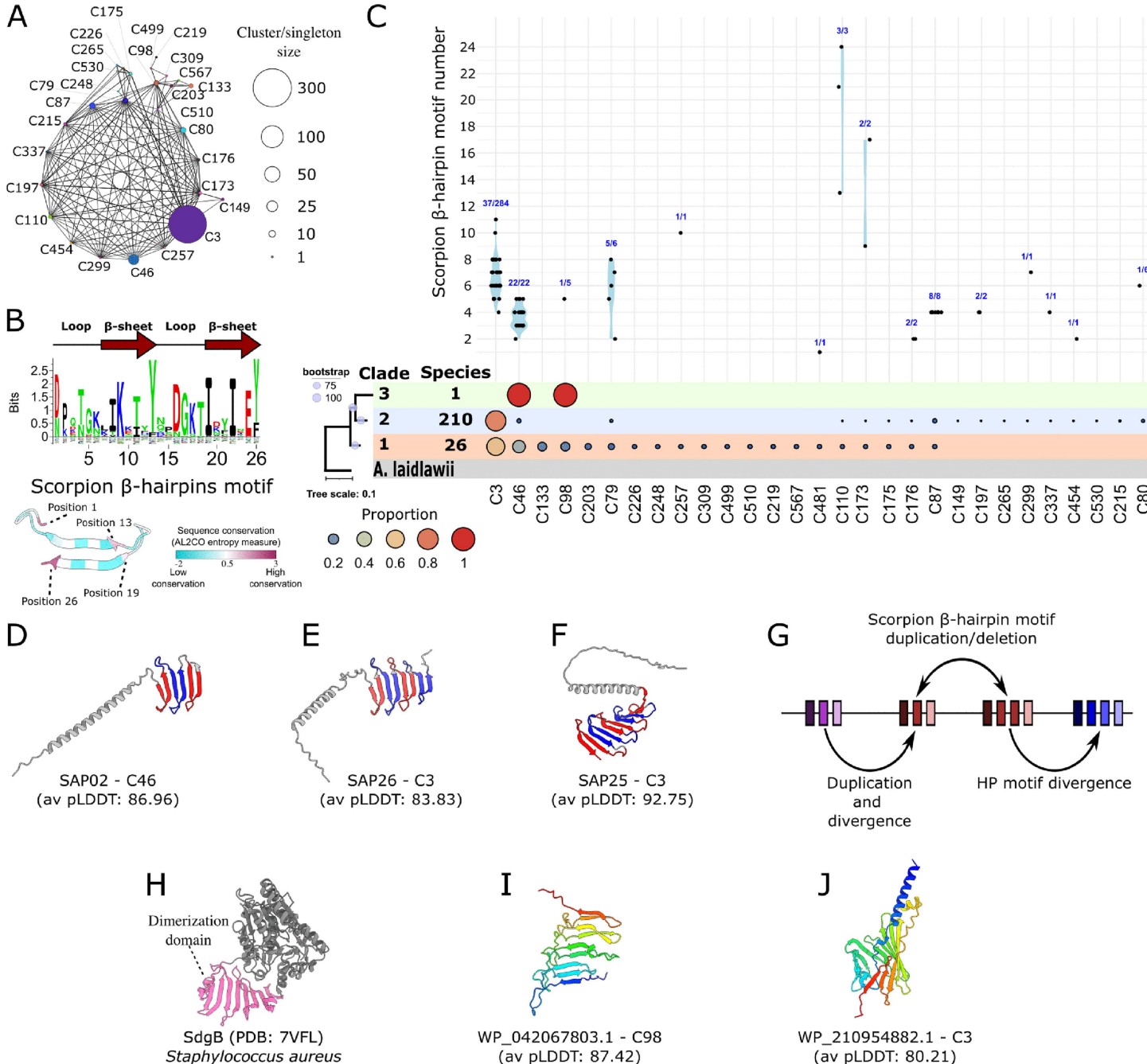

**Fig 4. β-hairpin motif repeats contribute to the diversity of PhAMEs with Scorpion-folds.** (**A**) Structure similarity network of PhAMEs with β-sheet fold. Structure-based clusters are indicated next to the nodes, with node size representing the number of PhAMEs within each cluster or singleton. Edges denote structural similarity between cluster members, as determined by Foldseek (E-value ≤ 0.001). (**B**) TOP: Level of residue conservation of the scorpion β-hairpin motif. Graph was generated using Weblogo3 and the secondary structure is represented across the top. Bottom: Predicted structure of β-hairpin motif with mapped residue conservation. Sequence conservation was calculated using the entropy-based method implemented in AL2CO [75] and visualized in ChimeraX. (**C**) Graph showing β-hairpin motif repeat numbers in clusters representing Scorpion PhAMEs in A. Relative number of PhAMEs in each cluster across the phytoplasma taxa and three major clades in the phytoplasma phylogeny (S1A Fig) are indicated as proportions below the graph. (**D-F**) Examples of secreted AY-WB phytoplasma proteins (SAPs) among the PhAMEs with predicted Scorpion-like folds. Tandem repeat units are highlighted in alternating blue and red. Cluster membership (as in panels A, B) and the average pLDDT score for each model are indicated. (**G**) Schematic overview of evolutionary processes that led to the amplification and diversification of proteins with Scorpion folds. (**H**) Crystal structure of

SdgB glycosyltransferase from *Staphylococcus aureus* (PDB: 7VFL) with the dimerization domain that has structural similarities to the β-hairpin domain of Scorpion PhAMEs shown in pink. (**I-J**) PhAMEs that have β-sheet structures but no obvious sequence similarities to the β-hairpin motif shown in (B). Cluster membership (as in panels A, B) and the average pLDDT score for each model are indicated. For (I-J), proteins are coloured from blue at the N-terminal end to red at the C terminus.

To explore the potential role of Scorpions, we performed a structure-based search against experimentally determined protein structures in the PDB using Foldseek. Interestingly, the scorpion β-sheet domain has structural similarities to multimerization domains of *Streptococcus* and *Staphylococcus* Glycosyltransferases, including the dimerization domains of SdgB (PDB: 7VFK) and SdgA (PDB: 7EC2), two glycosyl transferases from *Staphylococcus aureus* (Fig 4H) [78]. The structural similarities between these domains indicate that Scorpions might also bind proteins and sugars in phytoplasma hosts.

## Conserved strategies of host interaction among mollicutes

To investigate structural similarities between PhAMEs and proteins from other Mollicutes, we performed a structure-based screen using Foldseek (E-value ≤ 0.001) against the AlphaFold Database (AFDB) [79] focusing on the Mycoplasmatota phylum (NCBI Taxonomy ID: txid544448). This dataset comprises over 200,000 predicted protein structures from at least 1247 organisms (S10 Table), encompassing a range of bacterial lifestyles, from free-living organisms to obligate intracellular pathogens. Our analysis identified shared folds between phytoplasma PhAMEs and proteins from other Mollicutes (Figs 5A, S17, S11 Table). The pattern of presence/absence of PhAME-related proteins was in many cases incongruous with the species phylogeny suggesting pervasive gene-loss/gain and/or HGT. Interestingly, several folds found in phytoplasma effectors were not found in other Mollicutes (S17 Fig, S11 Table), suggesting distinct infection mechanism by this phytoplasmas.

Notably, structural homologs of PhAMEs were found in other mollicute pathogens, with many having predicted secreted or membrane localization (S12 table). Particularly proteins belonging to PhAME clusters C122, C310, C480 and, C485 shared domains with proteins of *Mycoplasma bovis*, *Mycoplasma mycoides*, and invertebrate-associated spiroplasmas, entomoplasmas and mesoplasmas, but were more rarely detected in free-living mollicutes, such as acholeplasmas (Figs 5A, S17, S11 Table). Closer structural analysis revealed that these proteins shared a conserved core composed of two/three β-sheets partially wrapped around an α-helix, often arranged in tandem repeats or fused with additional domains. Structural comparisons with experimentally determined proteins in the Protein data bank (PDB) revealed similarities to the D3 and D4 domains of internalin InlK (PDB: 4L3A) from another pathogen, *Listeria monocytogenes* (Fig 5B), classified as 'immunoglobulin-like with an additional helix' domains (hIg-like) due to their β-sheet and α-helix architecture [98].

In *L. monocytogenes*, InlK recruits the host Major Vault Protein to evade autophagy [99]. In mollicutes, hIg-like domains are also present in several known or putative membrane-associated proteins of phytoplasmas, such as AMP (Fig 5C), P38 (Fig 5D), and VmpA (Fig 5E), and spiroplasmas and mycoplasmas, such as ScARP3d (Fig 5F) of *Spiroplasma citri* and P40 (Fig 5G) of *Mycoplasma agalactiae*. These proteins are implicated in pathogenesis, particularly host cell attachment. For instance, P38, P40, and P89 are adhesin-like proteins containing a conserved Mycoplasma Adhesin Motif (MAM), essential for host interaction [54]. Our structural analysis shows that the MAM maps onto the hIg-like domain (S18A Fig), suggesting a direct role in host protein binding. This aligns with known experimental data, as VmpA, which contains four tandem hIg-like folds, facilitates phytoplasma adhesion to host cells, likely via interaction with a leucine-rich repeat protein, and promotes insect cell entry via clathrin-mediated endocytosis [100,101]. Phytoplasma AMP proteins have a transmembrane domain at their C-termini, in agreement with their localization on the exterior of phytoplasmas cells [102,103]. Similarly, the phytoplasma AMP protein, which contains two hIg repeats, interacts with actin and ATP synthase subunits within insect cells and plays a role in determining insect vector specificity during phytoplasma transmission [51,52], illustrating how conserved structural modules diversify to mediate interactions with components of different hosts.

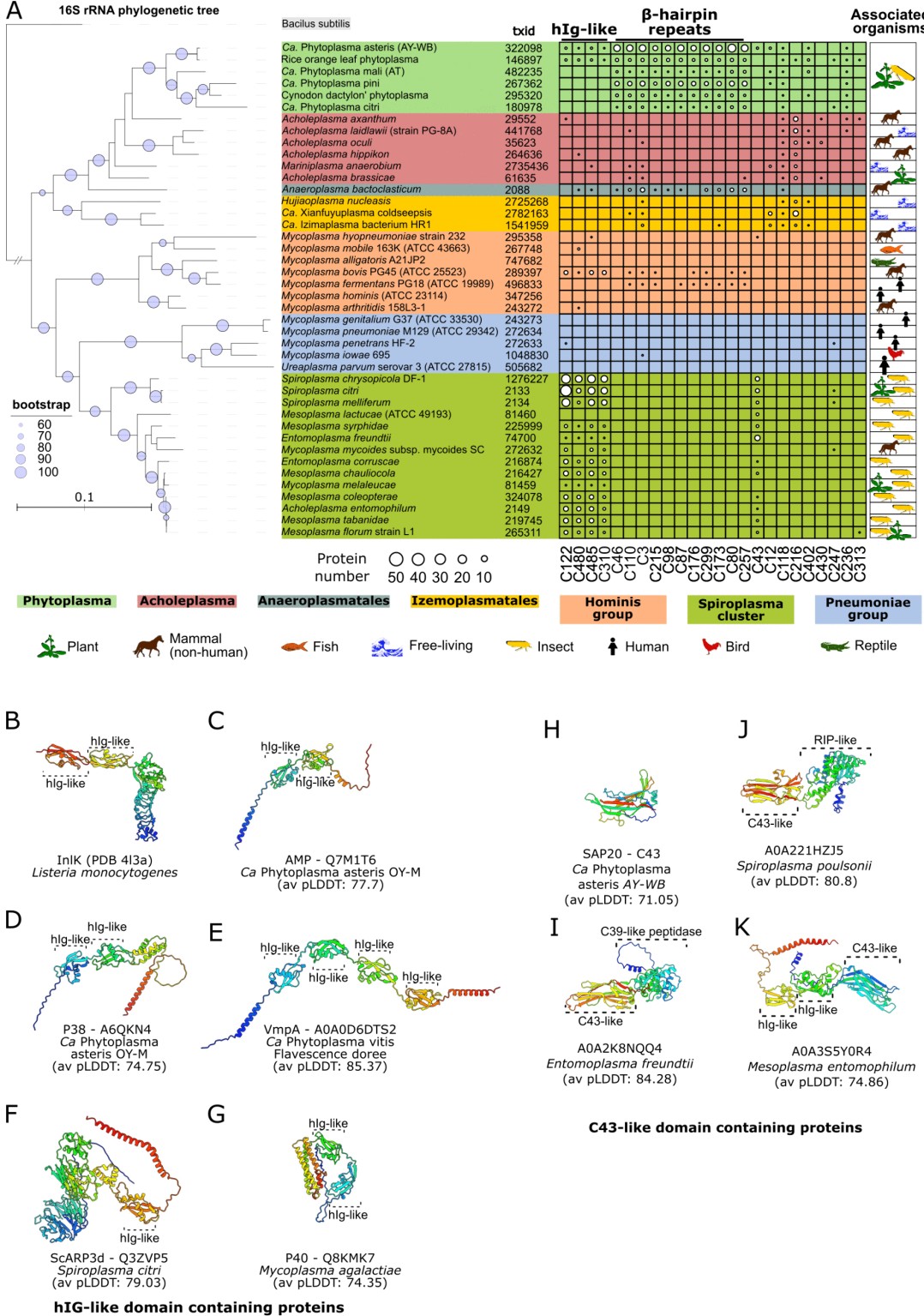

**Fig 5. Phytoplasma PhAMEs and other proteins conserved among Mollicutes. (A)** Phytoplasma PhAMEs exhibit structural folds similarly to proteins found in other bacteria of the class Mollicutes. Sizes of the white circles within the table indicate the number of proteins in each species that share a structural fold with PhAMEs present in one or more phytoplasmas. PhAME AF2 models were queried against a structural database built from the

AlphaFold Protein Structure Database [79], restricted to proteins from the phylum Mycoplasmatota (NCBI Taxonomy ID: txid544448), using Foldseek [80] (E-value ≤ 0.001). Six RePh are shown based on the high-level completeness of their proteome prediction. Only hits in which both proteins have an average pLDDT score ≥ 50 were included. Selected PhAME clusters are shown. The maximum likelihood phylogenetic tree shown on the left was constructed based on 16S rRNA gene sequences using the RAxML program with the GTRGAMMA model, following multiple sequence alignment with MUSCLE [81]. Node support was assessed with 500 bootstrap replicates, and bootstrap values ≥ 60% are indicated by purple circles. Branch lengths are proportional to nucleotide substitution rates (see scale bar). *Bacillus subtilis* was used as the outgroup. NCBI taxonomic identifier is indicated (txid). Tree visualisation was performed using the iTOL web server. The heatmap was generated using TBtools-II. Mollicute clades representing known genera are colour-coded, and species are annotated to indicate their host associations or free-living lifestyles [82–97]. *M. melaleucae* and *M. florum* were found on plant surfaces and are believed to be product of transport by insects. **(B)** Experimentally determined structure of InlK from the Gram-positive bacterium *Listeria monocytogenes* (PDB: 4L3A) showing immunoglobulin-like folds with an additional α-helix (hIG-like). **(C-G)** Other virulence proteins with hIG-like folds present in mollicute species. Average pLDDT scores are indicated. **(H)** Predicted structure of the AY-WB phytoplasma SAP20 effector that is a member of structure-based cluster C43. Average pLDDT score is indicated. **(I–K)** Selected proteins with C43-like fold from other mollicutes. Average pLDDT scores of each model are indicated. Due to the low average pLDDT score of A0A221HZJ5 in the AlphaFold Protein Structure Database, the structure model was re-predicted (as described in the Materials and Methods section). For B-K, proteins are coloured from blue at the N-terminal end to red at the C terminus.

Interestingly, the IMP protein from phytoplasmas adopts a different structure (PDB: 8J8Y), resembling the talin rod domain-containing protein 1 rather than the hIg-like motif [104] (S18B Fig), and binds plant actin [53,105], demonstrating independent evolutionary origins for actin binding in phytoplasmas.

Another fold shared between PhAMEs and other mollicutes is found in cluster C43 (Fig 5A). Proteins in this group, such as SAP20 and SAP43 from AY-WB, adopt a β-sandwich-like fold without additional domains (Fig 5H). In other mollicutes, this fold appears in predicted extracellular proteins from insect-infecting bacteria that are fused to Peptidase C39 (Fig 5I), RIP-like rRNA N-glycosylase domains (Fig 5J) and hIg-like domains (Fig 5K). Such domains are also found in effectors from spiroplasmas [106] and other pathogens [107], implicating them in host interactions. Notably, SAP20 and SAP43 are strongly upregulated in phytoplasmas colonizing plants compared to those colonizing insects [39], suggesting that phytoplasma C43-fold PhAMEs most likely modulate plant processes. In contrast, the *Spiroplasma* species encoding homologous proteins are restricted to invertebrate hosts, with the exception of *S. citri*, which is also an insect-vectored plant pathogen.

## Discussion

We systematically examined effector diversity across 239 phytoplasma genomes, using sequence and predicted-structural relationships to explore how these effectors evolve under the constraints of reduced genomes and host-dependent lifestyles. We found that phytoplasma effectors diversify primarily through structural variation within a limited set of conserved protein folds. Particularly, the SAP05 fold recurs across multiple effectors, exhibiting differences in binding surfaces and repeat architecture. Compact α-helical structures were frequently observed, likely reflecting their structural stability and functional adaptability. In addition, we identified a previously uncharacterized effector family - Scorpions - defined by an α-helix and tandem β-hairpin motifs. Finally, we uncovered conserved folds shared not only among phytoplasmas but also with effectors from spiroplasmas and other intracellular mollicutes. Collectively, these findings highlighted how the effectorome of obligate intracellular pathogens can generate functional and structural diversity.

We predicted the structures of PhAMEs using computational modelling and clustered them based on both structural and sequence similarities. Notably, 80.5% of structural clusters containing four or more PhAMEs included at least one member from the 24 representative phytoplasmas (RePh group), which span the three major clades of the phytoplasma phylogeny and have fully sequenced genomes. Our dataset also accurately reproduced the known structures of two effectors with available crystal structures, validating the protein structure predictions in our pipeline. In addition, the clusters included representatives of previously characterized effectors with known molecular functions. Finally, our analyses revealed both similarities to established protein folds and the presence of novel structural architectures among phytoplasma PhAMEs, showcasing the strength of the structural phylogenomics approach [108]. Collectively, these findings indicate that our methodology successfully captured most of the structural diversity within the broader phytoplasma pan-effectorome.

While some PhAME domains share similarities with proteins from other mollicutes, the overall effector repertoire of phytoplasmas appeared largely unique. For instance, a Foldseek search (E-value ≤ 0.001) against the clustered AlphaFold Protein Structure Database (AFDB50), which includes over 52.3 million predicted protein structures across the tree of life [62], identified no structural homologs for PhAMEs exhibiting SAP05-like folds. This underscored the distinctiveness of phytoplasma effectors at the structural level. Despite their reduced genomes and limited capacity for HGT, phytoplasmas appear to diversify their effectors through multiple mechanisms, including intragenic domain duplications, domain fusions with unrelated sequences, sequence variation at protein–protein interaction surfaces, and expansions of repeat units. These strategies likely enable functional diversification while operating within the genomic and evolutionary constraints of their obligate intracellular lifestyle.

Several phytoplasma effectors with known functions have convergently evolved a common strategy: targeting evolutionarily constrained domains of plant TFs and promoting their degradation through recruitment of the 26S proteasome machinery. By exploiting conserved and constrained interaction surfaces within TF families, these effectors achieve broad yet specific targeting. For example, SAP05 mimics double-stranded DNA to bind the DNA-binding domains of multiple TFs, whereas SAP54 and SAP11 imitate TF multimerization domains. These structural mimicry strategies enable phytoplasma effectors to destabilize multiple, often functionally redundant, host proteins, enhancing their impact despite possessing a limited effector repertoire.

SAP05 effectors represent a compelling new addition to the expanding class of DNA mimic proteins [109], which structurally resemble nucleic acids to hijack nucleic acid-binding proteins. Effectors from other pathogens, such as *Ralstonia* PopP2 and *Pseudomonas* AvrRps4, also target conserved DNA-binding surfaces of WRKY TFs [110,111]. However, rather than promoting degradation, PopP2 acetylates key lysine residues, blocking DNA binding and thus transcriptional activity [112]. While direct inhibition of DNA binding may be sufficient to suppress TF function, the extensive redundancy within plant TF families likely drives phytoplasmas to adopt degradation-based strategies, ensuring more complete suppression with fewer effector molecules. Consistently, plants expressing SAP05 and a mutant version of RPN10 that cannot bind SAP05 (38GA39>HS) do not exhibit witches' broom symptoms. This supports the conclusion that blocking the DNA-binding surface of TF alone is not sufficient for SAP05 activity [30]. SAP05 proteins reveal how reduced-genome pathogens use clever molecular mimicry to hijack host cell machinery.

The SAP05-like fold emerges as a central diversification platform. It features a conserved core with variable interaction loops, allowing partner specificity diversification, as observed in SAP05 and PM19_00185. We previously showed that mutations in the SAP05 loop surface disrupt interaction with TFs but not with the proteasomal ubiquitin receptor RPN10 [33]. Loop swapping experiments between homologues that bind distinct TFs (e.g., SPL vs. GATA) confirmed that loop composition dictates target specificity [33]. Here, we identified additional SAP05-like proteins with divergent loops and distinct partners, suggesting these loops are naturally evolvable. This offers opportunities for engineering SAP05 variants capable of selective ubiquitin-independent degradation by maintaining proteasomal binding while altering target specificity.

Our structural analyses revealed that SAP54, which adopts an antiparallel coiled-coil configuration [42], has two distinct, oppositely oriented binding surfaces: one for MTFs and another for the C-terminal ubiquitin-binding domain of RAD23. Other residues involved in the binding with MTFs are in the same surface as N59 [71], while those involved in the binding with RAD23 are on the opposite surface. These dual interfaces enable SAP54 to function as a molecular scaffold that bridges MADS-box TFs to RAD23, a component of the 26S proteasome, thereby mediating their targeted, ubiquitin-independent degradation [30,43]. This bipartite binding mechanism mirrors that of SAP05, which also recruits host TFs for degradation via direct proteasome engagement. Importantly, we demonstrated that SAP54 can be engineered to reduce TF binding while preserving RAD23 interaction, offering a potential route toward synthetic SAP05-like systems for selective, ubiquitin-independent protein degradation [33]. Collectively, these findings highlight a conserved phytoplasma effector strategy to hijack the host proteasomal machinery through structurally mediated dual interactions.

SAP11 and SAP54 operate distinct strategies of host protein targeting despite both adopting α-helical folds. SAP11 interacts with folded dimerization domains of TCPs using flexible helices and loops, while SAP54 employs a rigid, hydrophobically stabilized structure with two spatially separated surfaces to simultaneously engage MADS-box TFs and RAD23. These mechanisms illustrate a broader strategy in phytoplasma effector evolution: the mimicry of conserved protein–protein interaction interfaces to engage multiple members of expanded host protein families. The abundance of PhAMEs with simple α-helical folds underscores the central role of modular, protein-binding α-helical effectors in the phytoplasma toolkit.

We identified a novel effector family, which we named "Scorpions", characterized by an N-terminal coiled-coil and a C-terminal array of β-hairpin repeats. These proteins are structurally diverse, with variation in repeat number and sequence. Multiple Scorpion proteins with different repeat configurations often co-occur in the same phytoplasma, suggesting functional diversification. β-hairpin repeats are associated with target binding, subcellular localization, and structural stability [113], and they resemble β-sheet-mediated interaction domains found in dimerizing enzymes such as *S. aureus* SdgA/B [78]. We proposed that Scorpions function as molecular binders, with repeat architecture tuning interaction range and specificity. Notably, the use of 80% structural overlap as a clustering criterion may have split repeat-containing Scorpions into multiple clusters.

Our finding that phytoplasma effectors diversify primarily through structural variation within a limited set of conserved folds aligns with the localization of their genes within PMUs. These elements, which resemble conjugative transposons and carry IS3-family insertion sequences [22,27,114], range up to 20 kb in length, with shorter variants often enriched in pseudogenes [27]. PMUs are widespread across the phytoplasma phylogeny and contribute to genome size variation and gene duplication has previously been shown to drive effector sub-functionalization, as in the case of SAP05a and SAP05b from WBDL, which bind different transcription factor families [30]. The abundance of IS elements and pseudogenes in phytoplasma genomes mirrors patterns observed in other bacteria that have undergone severe population bottlenecks, where reduced selection efficacy facilitates the accumulation of mobile elements and pseudogenes [1]. These elements support adaptation to competitive host environments, especially under conditions of limited HGT. Isolation of phytoplasmas combined with selective pressures, may drive effector diversification, and when such effectors confer a fitness advantage, they are more likely to spread to other phytoplasmas when opportunities for HGT arise. HGT can occur under specific circumstances, as shown for mycoplasma integrative conjugative elements (ICEs) [21]. In phytoplasmas, PMUs were found to produce extrachromosomal units, particularly during insect vector colonization [22], suggesting a mechanism for PMU exchange during co-infection by multiple strains, which occurs [115]. Increasing evidence supports HGT of effector-containing PMUs among phytoplasmas [116,117]. Together, these findings highlight PMUs as dynamic reservoirs for the emergence, duplication, and diversification of novel effectors.

In contrast, genes encoding Scorpion-like effectors often reside on plasmids or in plasmid-derived chromosomal regions, as in *Ca*. Phytoplasma asteris AY-WB [27]. Since plasmid sequences are often excluded from genome assemblies, current estimates of Scorpion diversity are likely to be conservative. Interestingly, in *Ca*. Phytoplasma australiense, PMU sequences are integrated into plasmids [118], and recombination between plasmids and PMUs may explain hybrid PhAME architectures linking SAP05, Scorpion, and α-helical clusters. This suggests that genome instability, via PMU/plasmid interactions, may facilitate evolutionary innovation.

In conclusion, our study reveals that effectors targeting evolutionarily conserved and structurally constrained surfaces of host proteins are widespread among phytoplasmas. These effectors adopt compact and efficient folds with scaffolding functions, characterized by dual interaction surfaces that enable the linking of host proteins or integration of host pathways. Notably, PhAMEs with this mode of action have evolved multiple times independently—an outcome of convergent evolution driven by gene duplication, interface variation, domain fusion, and repeat expansion. These mechanisms have facilitated the emergence of novel folds and functions, despite the constraints of genome reduction and limited horizontal gene transfer. While a few effector folds are shared with other Mollicutes, the majority are unique to phytoplasmas,

highlighting their distinct evolutionary trajectory and refined strategies for host manipulation. Altogether, these insights deepen our understanding of phytoplasma biology and the principles governing pathogen–host interactions, while opening new avenues for the discovery and design of novel biomolecules with biotechnological potential.

## Materials and methods

### Sequence analysis

Mollicute nucleotide and protein sequences were downloaded from NCBI (https://www.ncbi.nlm.nih.gov/). 16S rRNA genes (≥ 97% complete) were aligned using MUSCLE [81] with default parameters. Maximum likelihood phylogenetic trees were constructed from the aligned 16S genes using the RAxML algorithm (GTRGAMMA model) [119] with 100 or 500 bootstrap in Geneious Prime (v. 2024.07). Effector protein sequences of mature SAP05, Phyl1/SAP54 and SAP11 from different phytoplasma species were aligned with MUSCLE and a maximum-likelihood phylogenetic tree was inferred with IQ-tree2 software [120], in the JIC High-Performance Computing Cluster, with 1000 bootstrap.

Phytoplasma genome completeness was evaluated with CheckM [57] v1.2.0 with the marker set for Mollicutes.

### Data mining for phytoplasma candidate effector proteins

Phytoplasma secretome was predicted as in Bai et al. [32], with small modifications. Briefly, phytoplasma protein sequences were downloaded from NCBI (S1 Table). Signal peptides within the N-terminal 20–70 amino acids of each protein were predicted using SignalP v3.0 with the following parameters: signalp -t gram + -f short -trunc 70 [121] with organism group set as gram-positive. Protein with HMM scores and cleavage site probability ≥ 0.50 were evaluated for transmembrane domains using TMHMM 2.0 [122] with the following parameters: tmhmm -short. Those peptides lacking transmembrane domains after removing the predicted signal peptide were identified as PhAMEs. Predicted mature PhAME sequences have been deposited in Zenodo [123].

### Sequence-based sub-clustering

Mature PhAME sequences were clustered using the MMseqs2 clustering algorithm [66] with the following parameters: --min-seq-id 0.5, --cov-mode 0, -c 0.8, and --cluster-mode 1.

### Protein structure prediction

The structures of mature PhAMEs from the 24 phytoplasmas, along with three representatives from non-represented sub-clusters, were predicted using AlphaFold2 [56,124] via ColabFold: AlphaFold with MMseqs2 BATCH notebook. Prediction parameters included msa_mode: MMseqs2 (UniRef+Environmental), num_models: 5, num_recycles: 12, and stop_at_score: 100. For each protein, the model with the highest average pLDDT score among the five generated was selected for further analysis, while proteins with an average pLDDT score below 50 were excluded from structural analysis. In cases where sub-clusters lacked structural models with an average pLDDT ≥ 50, three additional PhAMEs were selected from each sub-cluster for a second round of predictions. AlphaFold predicted structures of PhAMEs have been deposited in Zenodo [123].

### Structural clustering and similarity search

For each sub-cluster containing members with an average pLDDT > 50, the member with the highest average pLDDT score was selected as the sub-cluster representative. Representatives were then clustered using the Foldseek clustering algorithm [62] with the parameters -c 0.8, -e 0.001, --cov-mode 0, and --cluster-mode 1. PhAMEs were assigned to the same cluster as their sub-cluster representative.

Protein structural similarity was assessed using Foldseek [80] using the following parameters: foldseek easy-search -e inf --format-output "query, target, fident, alnlen, mismatch, gapopen, qstart, qend, tstart, tend, evalue, bits, alntmscore, qtmscore, ttmscore, lddt, lddtfull, prob" for proteins longer than 12 amino acids, with an all-versus-all comparison of predicted AlphaFold2 models with an average pLDDT ≥ 50. Low- and high-threshold similarity searches were performed using the parameters -e 0.1 and -e 0.001, respectively. Protein sequence similarity was evaluated with MMseqs2 [125], using the parameter -e 0.01. Network visualization and analysis were conducted in Cytoscape 3.10.2 [126].

For structure-based similarity search against non-phytoplasma mollicutes, Foldseek was run locally against predicted structures with average pLDDT ≥ 50 corresponding to Phylum Mycoplasmota (txid54448) available from Alphafold Protein Structure Database [70] using the same parameters as for the all-versus-all search.

Structural similarity search against experimentally determined proteins was done locally against PDB100 20240101 [62] database using Foldseek with the same parameters as above.

### Scorpion hairpin (hp) repeat identification

Structural models of DUF2963-containing proteins were analyzed using a combination of the structure-based repeat identification tool RepeatDB-lite 1 [76] and the sequence-based prediction tool HHrepID [77]. To determine the number of repeats, a repeat identified by both tools was selected as a query for a PSI-BLAST [127] search against phytoplasma proteins. The search was performed with a significance threshold of e = 0.005 over five iterations. Due to the repetitive nature of these sequences, only proteins with more than one hit were retained after each PSI-BLAST search. Positive hits were used to construct a hidden Markov model (HMM) using HMMER 3.4 (http://hmmer.org).

### Coiled-coil identification

Structural models were evaluated for coiled-coils using Socket2 server [128] with the following parameters: Packing_Cutoff 8.5, Helix_Extension 0. TM alignment scores were calculated using Foldseek search algorith with alntmscore

### Molecular graphics

Structural superimpositions and depictions of proteins and ligands were produced with the Chimera v.1.8. Structural comparisons were made with the MatchMaker [129] tools as implemented in the Chimera software [130].

### Visual representation

Heatmaps were prepared using TBtools II [131] and graphs were prepared using ggplot2 package of R software (version 4.4.1). Phylogenetic tree visualisation was performed using the iTOL web server [132].

### Functional and structural annotations

Functional annotation was performed with interproscan-5.22.6 [133] with the default parameters

### Cloning of sap54 mutants

sap54 mutants comprising the sequence of the 91-amino acid long mature peptide without the N-terminal secretion signal peptide (33 aa) were amplified by site-directed PCR mutagenesis using primers containing wild-type and mutated sequences and the template SAP54 from AY-WB [1]. The PCR fragments containing *attB1* and *attB2* recombination sites at their 5' and 3' ends, respectively were cloned into the donor vector pDONR221 (Invitrogen) using the BP Clonase II enzyme mix (Invitrogen) and subsequently introduced into the destination vector pDEST32 (Invitrogen) using LR Clonase II (Invitrogen) following the manufacturer's instruction. All construct sequences including the substitution mutations were verified.

## Cloning of RAD23C and RAD23D domains

AtRAD23C and AtRAD23D (MacLean et al., 2014, [41]) were used to amplify the UBL and UBA domains based on the prediction by InterPro (http://www.ebi.ac.uk/interpro/): RAD23C UBL domain (aa 1–88); RAD23C UBA1 domain (aa 182–242), RAD23C UBA2 domain (aa 353–419). The PCR fragments with flanking *attB1* and *attB2* recombination sites were cloned into pDONR221 (Invitrogen) using BP Clonase II (Invitrogen) and then into the destination vector pDEST22 (Invitrogen) using LR Clonase II (Invitrogen). The cloned domains were validated by Sanger sequencing.

## Yeast two-hybrid assays

Assays were performed in the ProQuest Two-Hybrid system (Invitrogen) using pDEST22 (containing the GAL4 activation domain (AD) and Trp1 nutritional selection marker gene) and pDEST32 (containing the GAL4 DNA-binding domain (BD) and Leu2 selectable marker gene) in strain MaV203. MADS-box TFs SOC1, SEP3, AP1 and RAD23 homologues C and D fused to the AD (pDEST22; MacLean et al., 2014, [41]) were assessed in combination with the mature SAP54/sap54 mutant peptides fused to the BD (pDEST32). Yeast transformations were performed according to Gietz and Woods (2006) [134]. Yeast colonies co-transformed with pDEST22 and pDEST32 derivatives were grown on synthetic defined medium (SD) agar containing 2% glucose and lacking Trp and Leu (FORMEDIUM). Protein-protein interactions were assayed by analysing growth on SD agar deficient in Trp, Leu and His (FORMEDIUM) supplemented with 20 mM, 25 mM, 60 mM and 75 mM 3-Amino-1,2,4-triazole (3-AT, Sigma) or deficient in Trp, Leu, His and Adenine supplemented with 10mM 3-AT. For each yeast strain, four independent colonies were assayed for interaction by spotting 7-µl aliquots of a 200-µl yeast colony suspension (i.e., one yeast colony resuspended in 200 µl sterile water).

## Supporting information

**S1 Fig. Phytoplasmas contain a variable number of PhAMEs.** (A) A maximum likelihood phylogenetic tree was constructed using 16S rRNA gene sequences of ≥ 98% completeness aligned with MUSCLE [81] and inferred with RAxML using the GTRGAMMA model and 100 bootstrap replicates. Tree representation was done with iTOL. Three accessions belonging to the 16Sr IX group (Pigeon pea witches'-broom group) and two unclassified phytoplasmas were not included in the tree. *Acholeplasma laidawii* 16S rRNA gene was used as outgroup. Bootstrap values ≥ 60% are indicated by purple circles. Branch lengths are proportional to the nucleotide substitution rate, see scale bar for reference. The phylogenetic tree highlights three phytoplasma groups, indicated by distinct colours, and the separate 16Sr groups indicated with Roman numbers. 'UP' indicates unclassified phytoplasmas of the 16Sr group. For branches containing more than one species, the number of species within each branch is specified in triangles. The triangle numbers include phytoplasmas with < 98% 16S rRNA gene sequence completeness, but for which the 16Sr group is known. The PhAMEs/genome for each species is shown to the right of the phylogenetic tree, with PhAME numbers from the initial 24 phytoplasmas shown in red and those belonging to the 215 in white (see Fig. 1A). For Aster Yellows Witches'-broom phytoplasma, 3 PhAMEs reported in Bai et al., 2006 [32] that were not identified by our analysis were included. (B) Positive correlation between the number of PhAMEs and both the total number of protein-coding genes (left) and genome size (right). Point colour represents the CheckM score using the Mollicute marker set, an estimate of genome completeness [57]. Genomes from representative phytoplasmas (RePh) are outlined in black. Pearson correlation coefficient (*r*) and p-value (p) are shown. (TIF)

**S2 Fig. Most PhAMEs share sequence similarity with effectors from representative phytoplasma (RePh) species.** Top: Number of PhAMEs grouped by sequence-based sub-cluster size. Green bars indicate PhAMEs that cluster with those from RePh species; yellow bars indicate PhAMEs that do not. Percentages above bars reflect each group's proportion of the total PhAME set. PhAMEs with mature lengths < 13 amino acids are included in the "PhAMEs in sub-clusters/ singletons without RePh" category. Bottom: Number of sub-clusters and singletons grouped by membership size. Salmon

area denotes sub-clusters containing at least one RePh-derived PhAME; cyan area denotes those without. Percentages below pie charts represent the proportion of sub-clusters in each category relative to the total.
(TIF)

**S3 Fig. Most PhAME structures are predicted at high confidence.** (A) Distribution of average pLDDT scores for AlphaFold v2.0-(AF2) predicted models of all PhAMEs and those from the RePh species subset only. Box plots represent the interquartile range (IQR), with whiskers indicating the full range (minimum to maximum). Individual outliers are shown as jittered points. (B) Average pLDDT scores of PhAMEs in each of the 24 representative phytoplasma (RePh) species. The three major phytoplasma clades and their corresponding 16Sr groups are indicated at the top of the graph.
(TIF)

**S4 Fig. AF2-predicted models match the experimentally determined crystal structures of the SAP05 and Phyl1/ SAP54 PhAMEs.** Structural alignment of experimentally determined structures of SAP05 from (A) AY-WB and (B) OY-M phytoplasmas and SAP54 from (C) OY-M and (D) PnWB phytoplasmas with their corresponding AlphaFold-2 models. RMSD scores below the aligned structures were calculated by the Matchmaker command from ChimeraX.
(TIF)

**S5 Fig. Most PhAMEs are grouped into structurally related clusters, supported by high-confidence structural models.** (A) Average pLDDT score distribution of AlphaFold-2-predicted models for sub-cluster representatives, where each representative is the sequence with the highest average pLDDT score in its sub-cluster. Box plots represent the interquartile range (IQR), with whiskers indicating the full range (minimum to maximum). (B) Proportion of confidence prediction categories across clusters and singletons ordered by the highest proportion of members falling into their most confident prediction category. X-axis labels indicate the number of clusters containing at least one member within the corresponding confidence category. For each sequence-distinct PhAME, only the highest confidence model is represented. (C) Number of sequence-similarity-based sub-clusters within structure-based clusters. Number of PhAMEs with membership to each cluster is indicated in colours. Singletons were not included. (D) Top: Number of PhAMEs assigned to structure-based clusters, grouped by cluster size. Salmon bars represent PhAMEs that cluster with those from RePh species, while dark blue bars represent PhAMEs that do not. Percentages above bars are calculated relatively to the total number of clustered PhAMEs. Percentages within bars are related to PhAMEs with assigned clusters. Bottom: Number of structure-based clusters or singletons, grouped by membership size. Blue area denotes clusters containing at least one RePh-derived PhAME; orange area denotes those without. Percentages below pie charts represent the proportion of clusters in each category relative to the total.
(TIF)

**S6 Fig. Predicted structures of PhAMEs belonging to different clusters with structural similarities.** (A) Structural prediction of SAP05 (left), with membership to cluster C41 and the structurally related SAP49 with membership to C41 (right), both from AY-WB phytoplasma. (B) Structural prediction of the AY-WB phytoplasma effector SAP54 (left), a member of the large cluster C9, which comprises numerous structurally similar effectors and is linked to other related clusters such as C48. Shown on the right is the predicted structure of MD3178133.1 from Sweet Potato Little Leaf Phytoplasma, a representative of cluster C48. (C) Predicted structures of AY-WB phytoplasma effectors SAP25 (cluster C3, left) and the structurally related SAP02 (cluster C46, right). Proteins are coloured from blue at the N-terminal end to red at the C terminus.
(TIF)

**S7 Fig. SAP05 homologues are structurally conserved.** (A) RAxML phylogenetic tree constructed using the maximum likelihood method with mature effector sequences from cluster 41, with 1000 bootstrap replications. Homologues

reported in Huang et al., 2021 that failed to be identified by our pipeline were included [30]. Bootstrap values ≥ 70% are indicated by purple circles. SAP05 homologs with experimentally validated abilities to mediate degradation of SPL (green), GATA (blue), or both families of TFs (orange) are indicated [30]. Branch lengths are proportional to the amino acid substitution rate, see scale bar for reference. Tree representation was done using iTOL web server [132]. (B) MUSCLE [81] protein sequence alignment of the effectors shown in the phylogenetic tree (panel A). Secondary structure is depicted on top of the alignment using the SAP05^AY-WB full length protein as a reference. Experimentally validated residues of SAP05^AY-WB involved in the interaction with TF or RPN10 are indicated below the consensus sequence of the alignment (black letters). Residues in the alignment are coloured based on their chemical properties. All SAP05 homologs clustered in subgroup sC42, except for one that clustered in sC535, as indicated. Duplicated sequences were included once. (C) Top: Superimposition of predicted structures from a select number of SAP05 homologs to the crystal structure of SAP05^AY-WB (PDB: 8PFC), performed using the Matchmaker command in ChimeraX. Structures are coloured by root mean square deviation (RMSD), with blue indicating high similarity and red indicating low similarity to SAP05^AY-WB, and regions with RMSD > 5 Å indicated in grey. Bottom: AF2-predicted structural models the same SAP05 homologs with the additional inclusion of MDV3198312.1. Coloured outlines indicate experimentally validated ability to mediate degradation of SPL (green), GATA (blue), or both transcription factor families (orange) [30]. (D) Level of conservation of residues among SAP05 homologs in the cC42 sub-cluster (excluding MDV3198312.1) mapped onto the SAP05 crystal structure. Sequence conservation was calculated using the entropy-based method AL2CO [75] implemented and visualized in ChimeraX. (E) Front (left) and top (right) views of *A. thaliana* GATA18 bound to SAP05^OY-M (PDB: 8J48). (F) Front (left) and top (right) views of *Homo sapiens* GATA3 (HsGATA3) binding to its target DNA sequence (GATA DNA BS) (PDB: 4HCA). (G-H) Overlapping GATA transcription factors surfaces interact with SAP05 and DNA. Structural superimposition of *A. thaliana* GATA18 in complex with SAP05 (PDB: 8J48) and *H. sapiens* GATA3 bound to DNA (PDB: 4HCA), performed using the Matchmaker command in ChimeraX (G). Close-up view of GATA surfaces that interact with SAP05 and DNA with amino acid residues involved in the interaction with both SAP05 and DNA represented as sticks and labelled (H). Grey spheres indicate $Zn^{2+}$ ions (E-H).
(TIF)

**S8 Fig. SAP05-like proteins share high structural similarity despite low sequence identities.** (A) Levels of structural (left) and sequence (right) similarity among proteins with SAP05-like folds in structure-related clusters. Sequence identity values were calculated based on structure-based alignments using Foldseek. For each sequence-distinct PhAME, only the highest confidence model is represented. (B-G) Parts of proteins with predicted SAP05-like folds from representatives of selected clusters shown in Figs. 2C–F and 2I. Colours indicate structural similarity based on RMSD relatively to the experimentally determined structure of SAP05^AY-WB, using the Matchmaker command in ChimeraX. Regions in grey indicate RMSD > 5 Å. The representative model with the highest average pLDDT score of each cluster was used for alignment. SAP05-like domains were defined via sequence-independent multiple structure alignment using FoldMason [67], with additional residues included to encompass the full β5 β-sheet. Cluster number and average pLDDT of full-length proteins are indicated below the structures.
(TIF)

**S9 Fig. SAP54/Phyl1 phylogeny.** (A) Maximum likelihood phylogenetic tree constructed from C9 cluster SAP54/Phyl1 effector alignment shown in (A) with 1,000 bootstrap replicates (IQ-TREE2, [120]). Bootstrap values ≥ 70% are indicated by purple circles. Branch lengths are proportional to the amino acid substitution rate (see scale bar). The tree was visualised using the iTOL web server [132]. Phylogenetic clades identified by Iwabuchi et al. (2020) [71] are colour-coded. (B) Crystal structures and AlphaFold-2 models of representative members of the C9 cluster, coloured from blue at the N-terminal end to red at the C terminus.
(TIF)

**S10 Fig. Predicted structural basis for the SEP3–SAP54–RAD23 complex.** (A) AlphaFold-Multimer (AF-M) prediction of the Phyl1/SAP54$^{OY-M}$–AtSEP3 complex. AF-M correctly identifies residues 146–175 of AtSEP3 (highlighted in blue) as involved in the interaction with SAP54, consistent with prior data [43]. Right: Predicted Aligned Error (PAE) plot for the highest-confidence model, indicating estimated positional uncertainty between residue pairs. Blue represents low predicted error (high confidence), red indicates high error (low confidence). (B–C) Surface representations of the predicted SAP54$^{OY-M}$/Phyl1–AtSEP3 complex, with Coulombic electrostatic potential mapped onto (B) SAP54$^{OY-M}$/Phyl1 and (C) AtSEP3. Surfaces are coloured from positive (blue) to negative (red) charge. Electrostatic potential was calculated using ChimeraX. (D) Crystal structure of the AtSEP3 K-domain dimer (PDB: 4OX0). Helix 1, involved in the dimerization and Helix 2, involved in the dimerization and tetramerization of SEP3 are indicated. (E) Structural superposition of the predicted SAP54$^{OY-M}$/Phyl1–AtSEP3 complex with the AtSEP3 K-domain dimer, highlighting overlapping interfaces involved in MADS-box multimerization and SAP54 binding. (F) AtSEP3 K domain with one monomer coloured as in (B). (G) Predicted SAP54$^{OY-M}$/Phyl1–AtRAD23C complex, generated using AF-M. The C-terminal UBA2 domain of RAD23C is correctly predicted to mediate the interaction, as supported by experimental data (Fig. 3A; [43]). (H) PAE plot of the highest-confidence SAP54$^{OY-M}$/Phyl1–UBA2(AtRAD23C) model, coloured as in (B). (I) Structural superimposition of the SAP54$^{OY-M}$/Phyl1–AtRAD23C complex using ChimeraX MatchMaker tool. (J) Crystal structure of the UBA domain of human NBR1 in complex with ubiquitin (PDB: 2MJ5). (K) Structural superposition of the predicted SAP54$^{OY-M}$/Phyl1–UBA2(AtRAD23C) complex with the UBA(NBR1)–ubiquitin complex, suggesting that SAP54 targets the ubiquitin-interacting surface of UBA domains.
(TIF)

**S11 Fig. Yeast transformation controls for Y2H assays related to Fig. 3E.** Growth on double dropout (SC-LW) medium confirms the presence of both constructs. EV: empty vector; AD: GAL4 activation domain; BD: GAL4 DNA-binding domain.
(TIF)

**S12 Fig. SAP54/Phyl1 effectors share conserved residues involved in interactions with plant targets.** (A) MUSCLE [81] protein sequence alignment of the C9 cluster SAP54/Phyl1 effectors. Secondary structure is depicted on top of the alignment using the SAP54$^{AY-WB}$ full length protein as a reference. Their corresponding phylogenetic clades [71] (Phyl-A, Phyl-B, Phyl-C, Phyl-D) are indicated in black on the right. Residues tested for their role in interactions with SAP54 binders (Fig. 3) are marked with triangles below the alignment (blue: RAD23 interaction; green: TF interaction). The I102 residue important for maintaining the SAP54/Phyl1 structure is indicated with a yellow triangle. (B) Predicted structure of SAP54$^{AY-WB}$, coloured from blue at the N-terminal end to red at the C terminus (top) residue conservation (middle) and Coulombic electrostatic potential (bottom), with the latter mapped onto the protein surface. Sequence conservation was calculated using the entropy-based method in AL2CO [75], and electrostatic potential was computed using the ChimeraX Coulombic command. Residues identified as affecting the interaction between SAP54 and its binders are highlighted.
(TIF)

**S13 Fig. Intragenic tandem duplication contributes to the diversification of α-helical folds.** (A) PhAME clusters with structural similarities to SAP54 proteins (cluster C9). Node sizes reflect the number of PhAMEs within the clusters. Edges represent pairwise similarities between clusters: grey edges indicate sequence similarities (MMseqs2, E-value ≤ 0.01), and orange edges indicate both structural and sequence similarities (Foldseek E-value ≤ 0.001, and MMseqs2 E-value ≤ 0.01). (B-G, I) Predicted AF2 structures of selected PhAMEs with membership to clusters indicated in (E). Cluster membership and average pLDDT structure confidence scores are shown. Proteins are coloured from blue at the N-terminal end to red at the C terminus. (H) The C48 cluster SAP54-like protein of Sweet Potato Little Leaf phytoplasma (SPLL) also interacts with *A. thaliana* RAD23 proteins. Yeast two-hybrid (Y2H) assays showing interactions of SAP54$^{AY-WB}$ and 'SAP54'$^{SPLL}$ with

*A. thaliana* RAD23 proteins. Right: Y2H assay to test interactions of SAP54[AY-WB] and *A. thaliana* RAD23. EV, empty vector control. AD, GAL4-activation domain. BD, GAL4-DNA binding domain. SD-LWH, triple dropout medium lacking leucine, tryptophan and histidine; 3-AT, 3-amino-1,2,4-triazole, a competitive inhibitor of the HIS3 enzyme. Growth on double drop-out (–LW) medium confirms the presence of both constructs. (J) Schematic overview of evolutionary processes that led to the amplification and diversification of proteins with SAP54-like folds.
(TIF)

**S14 Fig. PhAMEs consisting of simple α-helical folds contribute to the diversity of the phytoplasma pan-effectorome.** (A) Pie charts showing the proportions of clusters (left) and PhAMEs (right) with small α-helical fold. Single-tons were excluded from the analysis. (B) Structure similarity network of PhAMEs illustrating relationships among clusters, generated using a more relaxed threshold (Foldseek, E-value ≤ 0.1) than in Fig. 1B, to explore similarities among more divergent folds. Node size is proportional to the number of PhAMEs in each cluster. Clusters predicted to adopt helical structures are shown as triangles, while those with other folds are shown as circles. A single additional structural similarity identified under this relaxed threshold is highlighted in red. Unconnected nodes represent clusters with no detectable structural similarity to others. (C) Selected AY-WB phytoplasma effectors composed of antiparallel coiled-coil α-helices, belonging to distinct structural clusters. All effectors share coiled-coil regions predicted with Socket2, shown in black, with hydrophobic faces formed by leucine, valine, and isoleucine residues (shown in orange) oriented toward the protein interior. Cluster membership (as in panel B) and the average pLDDT score for each model are indicated. (D) Electrostatic surface potentials of antiparallel coiled-coil α-helix effectors involved in host target binding specificity differ among the effectors shown in panel C. Surfaces are coloured from positive (blue) to negative (red) charge. The positively charged surface of SAP54, involved in interaction with RAD23C, is highlighted. Electrostatic potentials were calculated using the ChimeraX Coulombic command.
(TIF)

**S15 Fig. Predicted structural features of SAP11 effectors involved in binding plant TCP transcription factors.** (A) The AF2-predicted structure of SAP11[AY-WB] structure indicating three α-helices (α) and two loops (L), coloured from blue at the N-terminal end to red at the C terminus. The average pLDDT score is indicated. (B) The AF-Multimer-predicted structure of the SAP11[AY-WB]-AtTCP10 complex. The model predicts that the TCP domain of the transcription factors mediates interaction with SAP11[AY-WB], while the SAP11[AY-WB] nuclear localization signal (NLS) and C-terminal region are not predicted to contribute to binding, consistent with experimental data [34–36]. Predicted Aligned Error (PAE) plot of the highest-confidence model, illustrating the estimated positional uncertainty between residue pairs. Blue indicates low predicted error (high confidence), and red indicates high predicted error (low confidence). (C) The pocket created by the two loops, α2 and parts of α1 and α3 creates a lipophilic area that binds the coiled-coil structure of the TCP domain involved in TCP dimerization – this TCP dimerization domain was preciously shown to be required for binding specificity to SAP11 proteins [35]. Surfaces are coloured from hydrophilic (dark cyan) to hydrophobic (gold). Lipophilicity potential was calculated using the ChimeraX mlp command. (D–E) Experimentally determined structure of a dimer of the coiled-coil region of the TCP10 domain involved in TCP10 dimerization (PDB: 7VPI). In (E), one monomer is shown with lipophilic potential mapped onto its surface representation. DNA binding region is indicated. (F) Structural superimposition of the AtTCP10 homodimer and the SAP11–AtTCP10 heterodimer shows that the helix-loop-helix motif of AtTCP10 adopts a similar fold when binding the effector as when interacting with other transcription factors.
(TIF)

**S16 Fig. The region that mediates binding to the coiled-coil dimerization domain of TCP transcription factors is conserved across SAP11 homologs.** (A) MUSCLE [81] protein sequence alignment of the effectors shown in panel A. Secondary structure elements are mapped above the alignment, based on their relative position in the SAP11[AY-WB] structure. Cluster membership (C8, C85, C445) is indicated on the right. Residues are coloured by hydrophobicity [135]. For

effectors with identical sequences a single representative was used for generating the alignment and further analyses. (B) RAxML phylogenetic tree constructed using the maximum likelihood method with mature SAP11 and SAP11-like effector sequences from clusters C8, C85, and C445, based on 1,000 bootstrap replicates. Bootstrap values ≥ 70% are indicated by purple circles. White text indicates homologs with confirmed TCP TF binding [73], while black text indicates homologs that have not yet been evaluated. The ability to bind Cyc/TB1, CIN, or Class I TCP transcription factors is indicated as per Correa Marrero et al., 2023 [73]. Phylogenetic clades as described in Correa Marrero et al., 2023 are colour-coded. Branch lengths represent amino acid substitution rates (see scale bar). Tree visualisation was performed using the iTOL web server [132]. (C) Structural predictions of selected SAP11-related cluster and sub-cluster members, showing a conserved architecture resembling SAP11^AY-WB. Cluster membership and average pLDDT scores for each model are indicated. Proteins are coloured from blue at the N-terminal end to red at the C terminus. (D) Sequence conservation (top) and hydrophilicity (bottom) of SAP11^AY-WB mapped onto its molecular surface. The hydrophobic interface predicted to mediate TCP binding displays high sequence conservation. Sequence conservation and molecular lipophilicity potential were calculated in ChimeraX using AL2CO entropy-based scoring and the mlp command.
(TIF)

**S17 Fig. Phytoplasma PhAMEs and other proteins conserved among Mollicutes – Extended from Fig. 5 to include phytoplasma-distinct folds.** Phytoplasma PhAMEs exhibit structural folds similar to proteins found in other bacteria of the class Mollicutes. Sizes of the white circles within the table indicate the number of proteins in each species that share a structural fold with PhAMEs present in one or more phytoplasmas. PhAME AF2 models were queried against a structural database built from the AlphaFold Protein Structure Database [79], restricted to proteins from the phylum Mycoplasmatota (NCBI Taxonomy ID: txid544448), using Foldseek [80] (E-value ≤ 0.001). Six RePh are shown based on the high-level completeness of their proteome prediction. Only hits in which both proteins have an average pLDDT score ≥ 50 were included. The maximum likelihood phylogenetic tree shown on the left was constructed based on 16S rRNA gene sequences using the RAxML program with the GTRGAMMA model, following multiple sequence alignment with MUSCLE [81]. Node support was assessed with 500 bootstrap replicates, and bootstrap values ≥ 60% are indicated by purple circles. Branch lengths are proportional to nucleotide substitution rates (see scale bar). *Bacillus subtilis* was used as the outgroup. Tree visualisation was performed using the iTOL web server [132]. The heatmap was generated using TBtools-II. Mollicute clades representing known genera are colour-coded, and species are annotated to indicate their host associations or free-living lifestyles [82–97]. *M. melaleucae* and *M. florum* were found on plant surfaces and are believed to be product of transport by insects. NCBI taxonomic identifier is indicated (txid). Cluster membership of characterized phytoplasma effectors is indicated.
(TIF)

**S18 Fig. The hIG-like domains contain Mollicute adhesin motifs (MAMs).** (A) MAMs mapped within the hIG-like domains (shown in red) of virulence proteins P38, P89, and P40. (B) A MAM was not detected in the experimentally determined structure of the *Ca*. Phytoplasma mali IMP protein (PDB: 8J8Y). IMP is coloured from blue at the N-terminal end to red at the C terminus.
(TIF)

**S1 Table. Phytoplasma genomes assembly quality used in this study.**
(XLSX)

**S2 Table. Prediction of putative Phytoplasma Effectors (PhAMEs).**
(XLSX)

**S3 Table. Phytoplasma proteins annotated as partial.**
(XLSX)

**S4 Table. Distribution of putative Phytoplasma Effectors (PhAMEs) by Structure-Based Clusters and Sequence-Based Sub-Clusters.**
(XLSX)

**S5 Table. InterPro signatures in PhAMEs.**
(XLSX)

**S6 Table. Phytoplasma AlphaFold-2 Structural Models from the AlphaFold Database (NCBI Taxonomy ID: txid33926) Exhibiting SAP05-like folds.**
(XLSX)

**S7 Table. SAP54 Mutants used in Fig. 3E and Supplemental Fig. 11.**
(XLSX)

**S8 Table. PhAMEs with Simple α-helical folds.**
(XLSX)

**S9 Table. Number of Scorpion hairpin motifs in PhAMEs.**
(XLSX)

**S10 Table. List of Mollicute AlphaFold-2 structural models in the AlphaFold database (NCBI Taxonomy ID: txid544448).**
(XLSX)

**S11 Table. Mollicute AlphaFold-2 Structural Models from the AlphaFold Database (NCBI Taxonomy ID: txid544448) Exhibiting Fold Similarity to PhAMEs.**
(XLSX)

**S12 Table. Secreted and Transmembrane Mollicute Exhibiting Fold Similarity to PhAMEs.**
(XLSX)

## Acknowledgements

We thank the JIC Informatics Platform and Norwich Biosciences Institutes Research Computing for installation of software, the Hogenhout lab members and and Pip J. Mountjoy for their critical reading and feedback and ChatGPT for improving the clarity of the text and code.

## Author contributions

**Conceptualization:** Federico G. Mirkin, Saskia A. Hogenhout.

**Data curation:** Federico G. Mirkin, Sam T. Mugford, Vera Thole, Mar Marzo, Saskia A. Hogenhout.

**Formal analysis:** Federico G. Mirkin, Sam T. Mugford, Vera Thole, Mar Marzo, Saskia A. Hogenhout.

**Funding acquisition:** Saskia A. Hogenhout.

**Investigation:** Federico G. Mirkin, Sam T. Mugford, Vera Thole, Mar Marzo.

**Methodology:** Federico G. Mirkin, Sam T. Mugford, Vera Thole, Mar Marzo, Saskia A. Hogenhout.

**Project administration:** Saskia A. Hogenhout.

**Resources:** Sam T. Mugford, Mar Marzo, Saskia A. Hogenhout.

**Software:** Federico G. Mirkin, Sam T. Mugford.

**Supervision:** Saskia A. Hogenhout.

**Validation:** Federico G. Mirkin, Mar Marzo, Saskia A. Hogenhout.

**Visualization:** Federico G. Mirkin, Saskia A. Hogenhout.

**Writing – original draft:** Federico G. Mirkin, Saskia A. Hogenhout.

**Writing – review & editing:** Federico G. Mirkin, Sam T. Mugford, Vera Thole, Mar Marzo, Saskia A. Hogenhout.

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
