## [Decision Letter · Decision Letter 0]

23 Sep 2025

PGENETICS-D-25-00866

Effector Innovation in Genome-Reduced Phytoplasmas and Other Host-Dependent Mollicutes

PLOS Genetics

Dear Dr. Hogenhout,

Thank you for submitting your manuscript to PLOS Genetics. After careful consideration, we feel that it has merit but does not fully meet PLOS Genetics's publication criteria as it currently stands. Therefore, we invite you to submit a revised version of the manuscript that addresses the points raised during the review process.

Please submit your revised manuscript within 30 days Oct 23 2025 11:59PM. If you will need more time than this to complete your revisions, please reply to this message or contact the journal office at plosgenetics@plos.org. Please include the following items when submitting your revised manuscript:

We look forward to receiving your revised manuscript.

Kind regards,

Frank O. Aylward

Academic Editor

PLOS Genetics

Sean Crosson

Section Editor

PLOS Genetics

Aimée Dudley

Editor-in-Chief

PLOS Genetics

Anne Goriely

Editor-in-Chief

PLOS Genetics

**Additional Editor Comments:**

Thank you for submitting your manuscript to PLOS Genetics. Your manuscript was assessed by two reviewers, who overall provided a favorable recommendation. Both reviewers have some helpful comments that could be incorporated in a new version to improve clarity and correct some minor issues. In particular, please check the issue noted by Reviewer 2 regarding the supplementary material.

**Journal Requirements:**

1) Please provide an Author Summary. This should appear in your manuscript between the Abstract (if applicable) and the Introduction, and should be 150-200 words long. The aim should be to make your findings accessible to a wide audience that includes both scientists and non-scientists. Sample summaries can be found on our website under Submission Guidelines:

https://journals.plos.org/plosgenetics/s/submission-guidelines#loc-parts-of-a-submission

3) Please ensure that the funders and grant numbers match between the Financial Disclosure field and the Funding Information tab in your submission form. Note that the funders must be provided in the same order in both places as well.

**Reviewers' comments:**

Reviewer's Responses to Questions

**Comments to the Authors:**

Reviewer #1: Markin et al. perform a systematic analysis to identify putative host-manipulating secreted effectors in the pangenome of phytoplasmas. The approach is rigorous and the results are informatively described by experts on the molecular biology of phytoplasma effectors. I think this is a high quality manuscript that will serve as a role model for similar systematic studies in other organisms.

##Minor comments / suggestions for improved clarity##

General—why “phytoplasma-associated” as part of the term for PhAMEs? Is this because “phytoplasma genomes” often contain host contigs due to their lack of culturability in axenic media?

L40 – grammar is awkward on “plant development, including witch’s broom and leaf-like flowers,”

General – Check the italics of taxonomic names like Wolbachia (L69) and Clostridium (L79), which are both genera. L352 candidatus. S. aureus in Figure 4H, and many instances in Figure 5. L623

L103-L104 – It would be helpful for non-phytoplasma experts if you indicate whether the Potential Mobile Units (PMUs) have been demonstrated to be bona fide MGEs or not. If the evidence is “looks like a duck, quacks like a duck”, I suggest saying “lie within apparent mobile genetic elements”.

L133 – Can you add a few words to clarify if “Membrane associated” means the proteins are associated with o embedded in the phytoplasma or the insect cell membranes?

L164 and in Table S1– can you indicate which markerset CheckM selected when the genomes were processed? In a different taxa, I have noticed that NCBI CheckM results are using a species-level marker set that seems to be biased (likely due to phylogenetic bias in the genomes used to define the set of single copy genes)

L166 – change “by all genomes” should be “by any of the genomes”; “by all genomes” means “core genes” in my opinion.

Figure S1b –I love your intuition to color the points based on this metric, but I found it a bit disappointing that so many genomes had “no data” for CheckM score. It’s a minor supplemental figure, so I don’t think you should necessarily re-run an analysis just to strengthen this figure.

L176 – I think you cited the wrong supplemental table. S5 Table seems to include the functional annotations. You might change “failed to identify domains” to “failed to identify functionally informative domains” because signal peptides count as a domain, they just don’t provide much information about function.

L179 – is “(sC)” a typo? I think you started using “subC” as an abbreviation.

L180 and L187 – Table S5 does not include the subcluster IDs. I think you should cite Table S4 here. Since I noticed two of these easy-to-make mistakes, I encourage you to carefully check each figure and table reference to make sure readers will find the information.

L224 “wondered of” should be “wondered if”. Google Docs’s spellcheck is good at catching this type of hard-to-notice typo.

L224-L237 – No critique major here, but I wanted to let you know that I appreciated your careful self-critique of your approach.

L475 – period is missing

L499 – This sentence says there were 2-24 scorpion-hp repeats, but Figure 4C indicates that many of the clusters had 0 of these repeats identified. I think the results sentence could be more clear.

L625 – “acetylates”, not “acetylate”

L809 – attB1 and attB2 should be italicized here. Also L819

A curiosity that I had as a non-expert: I wonder if the “population bottlenecks” are a good thing for phytoplasmas. Those transmission events might allow the populations of these bacteria to keep more orderly genomes than the Hodgkinia found in long-generation cidadas. https://www.pnas.org/doi/10.1073/pnas.1421386112

This was a very well-written manuscript. I want to thank you for pre-printing it so that I could ethically share it with my lab. It’s a role model for rigorous science and effective communication.

Reviewer #2: This work presents novel data and offers new insights about phytoplasma effectors. A great feature of this paper is that it addresses the whole effectorome of phytoplasmas, and thus is of interest for the whole phytoplasma research community (among others). A detailed structural basis for the activity of the already well-documented phytoplasma effectors SAP05, SAP54/PHYL1, SAP11 effectors is presented and stands as a complement to what was known already. New structural classes of yet uncharacterized effectors are unveiled. Authors bring evidence that "the overall effector repertoire of phytoplasmas appears largely unique".

The paper is well written, and provided data support the conclusions. Figures are rich and appealing.

Below is a listof several observations about discrepancies between text and supplemental data, or between text and figures, but none of them addresses the underlying approach of the work, or cast doubts on the conclusions drawn by the authors.

A surprising observation is that supplemental tables S2 and S4 appear to contain duplicate entries, and the authors seem to be unaware of it. As a result, several protein counts may be overestimated (details below). The conclusions of the paper are still standing however.

Here are my comments, sorted by order of appearance in the manuscript.

* p5 Section title. "Mining phytoplasma genome sequences identified 7162 PhAMEs". Table S4 lists the mentioned 7162 PhAMEs. However, analysis of table S4 showed that some entries are duplicated (for instance WP_213680437.1 is listed 20 times, and WP_004994795.1 19 times), likely due to the pooling of all phytoplasma proteins without removal of the duplicates. Therefore, the total number of PhAMEs as indicated in the title is overestimated (according to table S4, the total number of PhAMEs is rather 6362). Fortunately, this has no heavy consequence on the remaining of the analysis, thanks to the subsequent clustering step. However, the different counts indicated in the manuscript may be affected, as well as associated figures (Fig S2, etc. see also other comments below).

* p5 line 20 "phytoplasma-associated molecular effectors". In their manuscript, the authors designate the putatively secreted proteins as well as demonstrated effectors by the term "PhAME", defined as "phytoplasma-associated molecular effectors". I am not really convinced that this new term was required, as these are not new concepts or new biological objects, and I wonder why the authors did not simply referred to them by "phytoplasma effectors" (assuming that the term effector can be applied even if secretion or biological role has not been demonstrated).

* p6 line 4 Title "PhAMEs group into 553 clusters". It seems that in provided table S4, accession ABC65342.1 has no defined cluster (leading to a count of 552 regular clusters). Maybe this cluster was inadvertently deleted in the table.

* p6 line 5 "Functional annotation…" Table S4 is cited but should be Table S5 according to supplemental tables numbering. The same confusion is present in several places in the manuscript and in the table of contents of supplementary tables, so I believe that somehow the tables S4 and S5 have been exchanged in the supplemental tables file provided by the authors. I invite the authors to check the references to tables S4 and S5 throughout the manuscript, and/or correct the supplemental table file.

* p6 line 10 "1518 representative PhAMEs were selected". However, according to table S4, this number is 1524 (5638 entries with "Not_evaluated_pipeline" in Average_pLDDT column out of 7162). Part of the discrepancies may be explained by entries not associated with any cluster (subcluster "#N/A")?

* p6 line 17 "indicating that there is no obvious bias in the prediction quality of the secretomes". I am not sure whether this sentence refers to Fig S3A or S3B. I would suggest mentioning the figure referred to and reformulate the conclusion as it is not totally explicit.

* p7 line 5 "Of the 7255 identified PhAMEs…" This number is inconsistent with the 7162 PhAMEs announced earlier in the text, and also is different from the number observed in supplemental data provided by the authors (see comment on table S2).

* p7 line 7 Reference to table S5 is incorrect, should be table S4 (see above comment)

* p7 line 16 "We next wondered of…" -> change "of" to "if"?

* p8 line 15 "C3 and C36". C36 is not consistent with Supplemental fig S6C where C46 is mentioned instead.

* p8 line 31 "The loop surface exhibits greater sequence variability…" text is associated with Fig S7B but should be Fig S7D?

* p9 line 1 "The loop surface structure of SAP05…" this title appears to lack some words ("… recognized by DNA-binding interface....?) and should be corrected (compare with line 11).

* p9 line 28 "preducted" should be changed in "predicted"

* p10 section "Opposite facing surfaces of SAP54…". In this section, the authors use SAP54(OY-M)/PHYL1 in their structural models. However, in the next section ("Structural model validation…"), authors use SAP54(AYWB) for Y2H experiments. I guess that authors used the protein from strain OY-M, and not AYWB, for modelling because a crystal structure was available for OY-M (Iwabuchi et al. 2019). But this is not explained in the text. Moreover, a sentence (or reference) detailing how different the two proteins are could help to justify this choice.

* p11 lines 20 and 21. Text refers to fig S10M, which does not exist, and should be S10J. Same with S10N (S10K).

* p13 line 4 "S13M" should be replaced by S13I.

* p14 line 9. "SAP11-like proteins are distributed across clusters C8, C85, C582", whereas Fig S16A indicates C8, C85, C445. Moreover, cluster size has to be corrected (according to provided table S4, cluster C8 contains 154 entries). I let the authors check all cluster sizes in the manuscript, including associated figures.

* p14 line 26 "284 and 22 PhAMEs respectively". Due to the presence of duplicates in table S4, as already mentioned, correct number for cluster C3, according to table S4, should be 253. Text should be updated accordingly, but this may also have an impact on several associated figures of the manuscript.

* p15 line 10 "the scorpion beta-sheet domain has structural similarities to the dimerization domains of SdgB". Were there any other hits indicated by Foldseek when looking for structures similar to the scorpion domain?

* p15 line 23 / Fig 5A "Our analysis identified shared folds between phytoplasma PhAMEs and proteins from other Mollicutes". Apart from interesting examples of proteins involved in interaction with the host highlighted by the authors, it is not precised whether the other PhAMEs structural homologs detected in Mollicutes are associated or not with a putative secretion signal. It is possible that some of these folds are associated with cytosolic proteins, in mycoplasmas/spiroplasmas/acholeplasmas. But it is the same for phytoplasmas, consistently with the genomic scenarios presented by the authors (fig 2J, 4G). That is why I would appreciate that authors provide (as supplemental data) a heat map similar to fig 5A, but with the 24 representative RePh phytoplasmas for the phylogenetic tree, and if possible with the proportion of proteins predicted to have a signal peptide. This would allow to compare phytoplasmas and remaining Mollicutes on the same grounds, and possibly add further evidence for the genomic evolution/origin of PhAME folds suggested by authors. Comparison of folds shared by effectors and cytosolic proteins within phytoplasmas might also shed a bit of light on the origin of some effectors.

* p15 line 28 "PhAME clusters C122, C480, C605, C610 shared domains…". In fig 5A, clusters indicated to contain hIg-like folds are C122, C485, C480, C310.

* p21 line 16 "cleavage site probability > 50" -> presumably 0.50

* p22 line 13 "database containing available from": some words are missing.

* p22 line 32 "algorithm"

* Fig1C-J Please confirm that used colors only indicate relative distance to N/C terminus.

* Fig1B There is no arrow connected to SJP39 legend.

* Fig2A – Legend. Replace "yellow" by "cyan"?

* Fig3D There is one dashed line without legend (PHYL1/SAP54OY-M).

* Fig3E and Methods. Two different media for yeast growth are indicated : "SC – LWHA + 10 mM 3AT" and "SC – LWH + 40 mM 3AT". Although this is not a problem, as the authors do not directly compare the two subfigures, and K106E shows the same behavior on both media, I was wondering why different media were used and if this choice was dictated by affinity differences between tested proteins.

Moreover, the Methods section (p24) only mentions SD –LWH + 25/60/75 mM 3AT, thus is not consistent with media indicated in Fig3E.

* Fig5A The size of black circles in the row of "Phytoplasma" is somewhat misleading compared to the white circles, because most clusters indicated below the heat map have only few members (sometimes only 1). Therefore, it would be more informative to have an idea of the size of the cluster/number of proteins involved, and would substantiate the sentence "the overall effector repertoire appeared largely unique" in discussion. Please see also my related suggestion where I propose to indicate PhAMEs as well as structural homologues not predicted to be secreted ("p15 line 23 / Fig 5A").

Fig5B-G The names of bacterial species should be written with italics. Also, some phytoplasmas have poor designation : "OY-M" and "AY-WB" indicate strains only. "Phytoplasma dorée" for sure refers to flavescence dorée phytoplasma, but is not a correct formulation. Also the accession number for VmpA (fig 5E) seems strange ("ostparc").

* Fig5 legend. "M. malelucae" -> "M. melaleucae".

* Supplementary data. Submitted work relies on a large number of predicted structures currently not provided as supplemental data. I suggest that authors also consider releasing the predicted structures as supplemental data, if possible given the technical constraints. Note: I realized that authors already indicated in the submission form that they were willing to provide all of the predicted structures.

* S1 table. In the manuscript page 5, 1st paragraphe of "Results" section, the S1 table is indicated to contain the list of RePh genomes and the other group of 215 genomes. However, the S1 table does not contain data that indicate which accessions belong to the RePh group. The list of RePh accessions is available from fig S3B or from table S4 with slight effort, so I suggest to also include this information in table S1. Also, table S1 contains 238 accessions, whereas the text mentions 239 genomes.

S2 table. The S2 table contains 119567 entries. This number is not consistent with the 107543 proteins mentioned page 5 for the number of full-length proteins. Even if we remove the 13007 proteins annotated as partial and listed in table S3, we do not obtain the number of proteins mentioned in the text.

This table also contain redundant entries (106683 unique IDs).

Still in table S2, a discrepancy is also observed for the number of PhAMEs: 7162 according to the text, and 7747 according to table S2.

Another problem with table S2 is that some proteins are associated with some RefSeq (GCF) GenBank assemblies, whereas according to table S1 the full genome set contains only GCA assemblies. Tables S1 and S2 should be consistent with each other and with other figures (S3B for instance for RePh assemblies).

* S4 Table. Table S4 contains 7162 PhAMEs entries, as mentioned in the text. However, 2 of its entries (WP_011412484.1 and WP_041639883.1) appear in table S2 with no analysis by TMHMM or no prediction of signal peptide, which is contradictory to the definition of PhAMEs. Moreover, as already mentioned, table S4 contains 252 duplicated entries.

* Figure S6 – legend – (A) SAP49 is a member of cluster C28 according to text and figure, not C41 as mentioned in legend.

* Figure S7D –legend- subcluster cC42 is mentioned, whereas it should read "sC42".

* Figure S8A – On figure S8A, cluster C41 is associated with 8 rows/columns, and C55 with 7 rows/columns. However, according to table S4, cluster C41 contains 9 entries with structure predictions, and cluster C28 contains 8 entries. I have not checked the other clusters of fig S8A, but I wondered why not all of available structure predictions were represented on S8A fig? Also I suggest that authors indicate in the legend the meaning of "PPSP".

* Fig S14C It seems that "Alifatic" is usually written as "Aliphatic"

* Fig S15C (legend) "preciously" -> "previously"

* Fig S15D (legend) "In (E) one monomer is shown…"-> in (D)

* Fig S15E legend "TCP domain" in blue is incomplete

* Fig S16D "Lipophylic" shoud be written "Lipophilic"

**Have all data underlying the figures and results presented in the manuscript been provided?**

Reviewer #1: Yes

Reviewer #2: Yes

PLOS authors have the option to publish the peer review history of their article (what does this mean? ). If published, this will include your full peer review and any attached files.

**Do you want your identity to be public for this peer review?** For information about this choice, including consent withdrawal, please see our Privacy Policy .

Reviewer #1: No

Reviewer #2: **Yes: ** Christophe Garcion

**Figure resubmission:**
---

## [Editor Report · Decision Letter 1]

31 Oct 2025

Dear Dr Hogenhout,

We are pleased to inform you that your manuscript entitled "Effector Innovation in Genome-Reduced Phytoplasmas and Other Host-Dependent Mollicutes" has been editorially accepted for publication in PLOS Genetics. Congratulations!

Yours sincerely,

Frank O. Aylward

Academic Editor

PLOS Genetics

Sean Crosson

Section Editor

PLOS Genetics

Aimée Dudley

Editor-in-Chief

PLOS Genetics

Anne Goriely

Editor-in-Chief

PLOS Genetics

BlueSky: @plos.bsky.social

Comments from the reviewers (if applicable):

Thank you for submitting your manuscript to PLOS Genetics.

**Data Deposition**

http://datadryad.org/submit?journalID=pgenetics&manu=PGENETICS-D-25-00866R1

**Press Queries**

---

## [Editor Report · Acceptance letter]

PGENETICS-D-25-00866R1

Effector Innovation in Genome-Reduced Phytoplasmas and Other Host-Dependent Mollicutes

Dear Dr Hogenhout,

We are pleased to inform you that your manuscript entitled " 

Effector Innovation in Genome-Reduced Phytoplasmas and Other Host-Dependent Mollicutes" has been formally accepted for publication in PLOS Genetics! Your manuscript is now with our production department and you will be notified of the publication date in due course.

With kind regards,

Zsofia Freund

PLOS Genetics

On behalf of:
